



# Deep learning applied to $CO_2$ power plant emissions quantification using simulated satellite images

Joffrey Dumont Le Brazidec[1], Pierre Vanderbecken[1], Alban Farchi[1], Grégoire Broquet[2], Gerrit Kuhlmann[3], and Marc Bocquet[1]

[1]CEREA, École des Ponts and EDF R&D, Île-de-France, France
[2]Laboratoire des Sciences du Climat et de l'Environnement, LSCE/IPSL, CEA-CNRS-UVSQ, Université Paris-Saclay, 91198 Gif-sur-Yvette, France
[3]Swiss Federal Laboratories for Materials Science and Technology (Empa), Dübendorf, Switerzland

**Correspondence:** Joffrey Dumont Le Brazidec (joffrey.dumont@enpc.fr)

**Abstract.**

The quantification of emissions of greenhouse gases and air pollutants through the inversion of plumes in satellite images remains a complex problem that current methods can only assess with significant uncertainties. The anticipated launch of the $CO_2$M satellite constellation in 2026 is expected to provide high-resolution images of $CO_2$ column-averaged mole fractions

($XCO_2$), opening up new possibilities. However, the inversion of future $CO_2$ plumes from $CO_2$M will encounter various obstacles. A challenge is the $CO_2$ plume low signal-to-noise ratio, due to the variability of the background and instrumental errors in satellite measurements. Moreover, uncertainties in the transport and dispersion processes further complicate the inversion task.

To address these challenges, deep learning techniques, such as neural networks, offer promising solutions for retrieving

emissions from plumes in $XCO_2$ images. Deep learning models can be trained to identify emissions from plume dynamics simulated using a transport model. It then becomes possible to extract relevant information from new plumes and predict their emissions.

In this paper, we employ convolutional neural networks (CNN) to estimate the emission fluxes from a plume in a pseudo $XCO_2$ image. Our dataset used to train and test such methods includes pseudo images based on simulations of hourly $XCO_2$,

$NO_2$ and wind fields near various power plants in Eastern Germany, tracing plumes from anthropogenic and biogenic sources. CNN models are trained to predict emissions from three power plants that exhibit diverse characteristics. The power plants used to assess the deep learning model's performance are not used to train the model. We find that the CNN model outperforms state of the art plume inversion approaches, achieving highly accurate results with an absolute error about half of that of the cross-sectional flux method. Furthermore, we show that our estimations are only slightly affected by the absence of $NO_2$ fields

or a detection mechanism as additional information. Finally, interpretability techniques applied to our models confirm that the CNN automatically learns to identify the $XCO_2$ plume and to assess emissions from the plume concentrations. These promising results suggest a high potential of CNNs in estimating local $CO_2$ emissions from satellite images.



## 1 Introduction

The burning of fossil fuels, such as coal and oil, in power plants (PP), is a primary source of anthropogenic $CO_2$ emissions. Approximately $50\%$ of worldwide fossil fuel $CO_2$ emissions originate from large facilities, which encompass PPs (IEA, 2019; Nassar et al., 2022). As a result, maintaining regular monitoring of these emissions and possessing the capacity to control their reporting is crucial.

Observations from satellites like OCO-2 provide valuable data that can be utilised to estimate $CO_2$ emissions (Nassar et al., 2017; Reuter et al., 2019; Chevallier et al., 2019; Wu et al., 2020; Zheng et al., 2020; Nassar et al., 2022; Chevallier et al., 2022). Specifically, observations of $CO_2$ plumes, such as the plume transects obtained from OCO2-2 and OCO-3 satellites, offer a direct means of quantifying their source. The upcoming launch of the $CO_2M$ satellites in 2026 is anticipated to capture high-resolution images of $CO_2$ column-averaged mole fractions ($XCO_2$), further advancing our capabilities in this area. Leveraging these images, however, will present significant challenges (Wang et al., 2020).

$CO_2$ plumes are notoriously difficult to invert due to various factors, including image integrity issues caused by cloud cover or satellite overpasses, which result in missing data in the images used for analysis. Additionally, the estimation of the emissions associated with a plume is further complicated by the measurements low signal-to-noise (SNR) ratio. The noise component encompasses variations in the background as well as errors in the satellite measurements. The SNR problem stands as the main hurdle in the detection of the plume, a crucial step for inversion. Recent research conducted by Dumont Le Brazidec et al. (2022) has illustrated the remarkable ability of Convolutional Neural Networks (CNNs) to effectively overcome this obstacle. Lastly, another challenge stems from the uncertainties in the transport and dispersion processes, specifically, when it comes to estimating the effective wind driving the plume and determining its shape (Kuhlmann et al., 2019).

This paper addresses the second and third problems by employing deep learning techniques to perform inverse modelling of $CO_2$ plumes. In particular, we focus on developing techniques for inverting $CO_2$ plumes from PPs of different emission levels.

To assert the effectiveness of the method, the predictions of the deep learning model are compared against a state-of-the-art technique, the cross-sectional flux (CSF) approach. Plume inversion methods include approaches that use an atmospheric transport model to simulate the plume and compare it to observation (e.g., Pillai et al. (2016); Broquet et al. (2018)). They also include techniques that quantify emissions from a hotspot based on plume detection in satellite observations (Koene et al., 2021). These methods can be based on time-averaged plumes, such as the divergence method (Beirle et al., 2019; Hakkarainen et al., 2022), or on instantaneous images. Varon et al. (2018) compared several of these approaches, namely the Gaussian plume inversion, the integrated mass enhancement and the CSF method. Among these approaches, the CSF method is regularly diagnosed as being among the most robust and viable options (Varon et al., 2018; Koene et al., 2021), although recent developments suggest that alternative methods such as light cross-sectional flux or based on Gaussian plume may yield superior performance (Hakkarainen et al., 2023).

In this paper, the proposed plume inversion approach is based on convolutional neural networks. Here, plume inversion involves the analysis of an image to extract scalar or vector emissions data at different time steps. Therefore, this task can be framed as an image regression problem, where relying on CNNs can offer significant advantages (Chollet, 2017). CNNs, a





type of supervised learning method, can be trained on a comprehensive dataset, where all input variables (images) and associated output variables (like emissions) are known. Once trained, these CNNs can effectively process and draw conclusions from unseen observational imagery. CNNs employ convolutional layers to extract essential features from images. Each filter is

automatically trained to detect specific patterns, such as edges, corners, or other shapes within the image. By stacking multiple convolutional layers, CNNs become capable of learning intricate patterns, enabling them to capture increasingly complex features. The ability of CNNs to capture and learn spatial features in images makes them a popular choice for various image-related tasks, including image recognition, classification, and regression. Given the nature of our plume inversion task, they are particularly well-suited due to their ability to identify spatial features in images, such as plume shapes or intensity, that correspond

to specific emissions. This feature extraction approach effectively harnesses the knowledge embedded in transport models, enabling this automatic capture of plumes dynamics. This ability to capture such features has already been demonstrated by Dumont Le Brazidec et al. (2022), which study the segmentation of plumes in $CO_2$ images.

To train and test the CNN models, this paper relies on a synthetic dataset as $CO_2$M data will not be available until 2026. This dataset has been designed to possess similar key features as the forthcoming $CO_2$M satellite, such as resolution and the

availability of $NO_2$ data. This study specifically focuses on clear-sky conditions, and the influence of clouds is not considered in the analysis.

Before introducing the inversion methodology and results, we briefly describe the physical fields used to train and evaluate the CNNs. This includes the presentation of the simulated satellite fields in section 2 and of the model used to produce segmentation masks of the plumes used as inputs of the inversion model in section 3. The inversion methodology using CNN is

described in section 4, specifying the problem statement, the model, the training process, and the alternative method employed for comparison. The subsequent section 5 delves into the application of the model to three specific PPs. In particular, subsection 5.4 places special emphasis on the interpretability of the trained CNNs. Discussions on the limitations and future directions of this study can be found in section 6, while the conclusions are outlined in section 7.

## 2 Dataset of XCO$_2$, winds and NO$_2$ images

XCO$_2$ and NO$_2$ images are taken from the concentration fields simulated for the SMARTCARB project (Brunner et al., 2019). Using the COSMO-GHG model, the SMARTCARB simulations were performed in a region centred around Berlin and covered several nearby PPs. These simulations were used to produce synthetic observations of $CO_2$M and to evaluate various plume detection and inversion approaches (Kuhlmann et al., 2019, 2020, 2021; Hakkarainen et al., 2021, 2023). The data are hourly and cover an entire year. Their spatial resolution is $0.01°$ and sixty vertical layers ranging spanning from an altitude of $0$ to

$24\,km$ are used. More information can be found in Dumont Le Brazidec et al. (2022), which presented more extensively a dataset very similar to the one used in this study.

Images used to train and evaluate the CNN inversion models consist of $64 \times 64$ pixels, with each pixel covering an area of $2 \times 2\,km^2$. Each image is extracted from the SMARTCARB COSMO-GHG simulated fields so that one hotspot is located in the centre. In addition, the selected size guarantees the inclusion of the majority of the central hotspot plume within the image. The

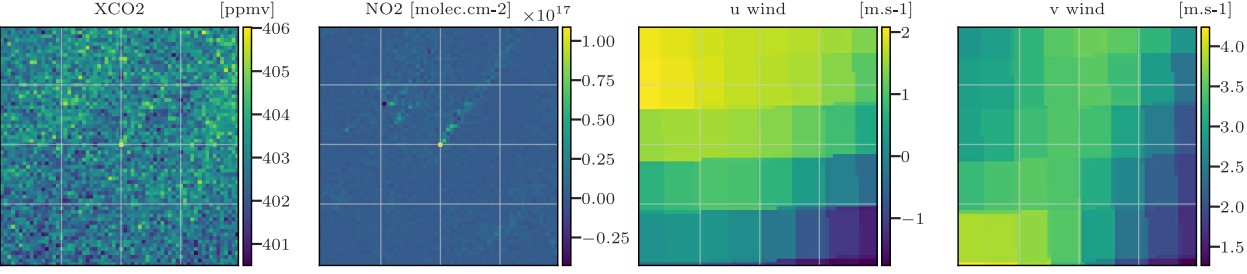

**Figure 1.** Examples of inputs used by the CNN model. The first, second, third and fourth columns represent the $XCO_2$ images, $NO_2$ images, vertically averaged $u$ and $v$ winds, respectively.

mapping from the original SMARTCARB fields resolution to the $2\,km$ resolution is performed by cubic spline interpolation (Virtanen et al., 2020). The $2\,km$ resolution was chosen to be consistent with the resolution expected for $CO_2M$ observations.

It is necessary to take into account the expected noise associated with the satellite instruments. For this, a Gaussian random noise of standard deviation $0.7\,ppm$, characteristic of the $CO_2M$ (Meijer, 2020), is added to the $XCO_2$ images.

In addition to $XCO_2$, ancillary data can be used to assist in the inversion of $XCO_2$ plumes. Considering that $CO_2M$ will provide measurements of $NO_2$ and the observed strong correlation between $NO_2$ and $CO_2$ plumes, noisy $NO_2$ fields are used in this study. The noise associated with a $NO_2$ field is implemented as the standard normal distribution multiplied by the $NO_2$ field values. The median standard value for $NO_2$ fields surpasses $1e15\,molec.cm^{-2}$, leading to an average noise level in the $NO_2$ field that exceeds the $CO_2M$ $NO_2$ requirement (less than $1e15\,molec.cm^{-2}$). Furthermore, ERA5 winds are used: the original resolution of $28\,km$ is mapped to $2\,km$ to be consistent with the $CO_2$ and $NO_2$ images. To overcome the circular data limitation of statistical methods, the $u$ and $v$ wind fields are used instead of the direction and magnitude components. This limitation corresponds to the statistical model's inability to correctly interpret wind directions where the value of 360 degrees is equivalent to 0 degrees. More precisely, we use 2D $u$ and $v$ wind fields which are calculated as the average of the zonal and meridional wind fields over the 37 lower ERA5 vertical levels, respectively, which corresponds roughly to the lowest $4000\,km$ of the atmosphere. Figure 1 presents a series of potential inputs to the CNN.

This paper only addresses the retrieval of PP emissions, although the training dataset includes the city of Berlin. More precisely, depending on the PP evaluated, the training dataset might be composed of any hotspot in {Berlin, Jänschwalde, Schwarze Pumpe, Boxberg, Turow, Patnow, Lippendorf, Opole, Dolna Odra}. The primary rationale behind prioritising the training of the model on PPs is the scarcity of cities in the dataset, which poses a challenge for the model to effectively learn and generalise for cities. However, Berlin is included in the training dataset as supplementary data to aid the model in its learning process. In the SMARTCARB dataset, the modelling of anthropogenic emissions, incorporating fixed diurnal, weekly, and seasonal cycles, was performed using the TNO-MACC III inventory (Kuenen et al., 2014). The emissions range, mean,



| Hotspot | min | max | mean | std |
|---|---|---|---|---|
| Berlin | 4.8 | 34.7 | 16.8 | 7.2 |
| Jänschwalde | 16.4 | 52.7 | 33.3 | 7.7 |
| Boxberg | 9.4 | 30.1 | 19.0 | 4.4 |
| Lippendorf | 7.5 | 24.1 | 15.2 | 3.5 |
| Turow | 4.3 | 13.8 | 8.7 | 2.0 |
| Schwarze Pumpe | 4.0 | 13.0 | 8.2 | 1.9 |
| Dolna Odra | 3.7 | 12.5 | 7.9 | 1.9 |
| Opole | 3.5 | 11.8 | 7.5 | 1.8 |
| Patnow | 2.9 | 9.2 | 5.8 | 1.3 |

**Table 1.** Emission statistics for the considered PPs and the city of Berlin. Fluxes are in Mt $CO_2$ /yr.

and standard deviation of each hotspot are given in Table 1. Moreover, data augmentation techniques are employed to expand the database, as detailed in section 4.2.1.

## 3 Application of the segmentation model

Utilising a segmentation algorithm to incorporate plume contours as critical prior information in plume inversion may yield significant benefits. In this section, we provide a brief description and application of the CNN-based method developed in Dumont Le Brazidec et al. (2022) that predicts plume contours in $XCO_2$ images. The methodology of Dumont Le Brazidec et al. (2022) involves employing an image-to-image U-net model, which generates images that are subsequently used as inputs for the CNN inversion model, as outlined in section 4.

Apart from a few specific points, the training and model choices are similar to those of Dumont Le Brazidec et al. (2022). A simpler encoder, with fewer neurons, is chosen since the $NO_2$ fields are used as inputs to the CNN. This simplification of the problem reduces the need for a complex encoder. In addition to $NO_2$ and $XCO_2$ fields, winds are also used to assist in the $XCO_2$ plume contour prediction, although experiments show that the addition of this data has very little influence on the predictions. Finally, the U-net models were designed to make predictions beyond the geographical region of their training data.

Specifically, the model that learns to predict the mask of the Boxberg PP plume from an image centred at Boxberg PP is trained on a dataset excluding the images centred at Boxberg.

In Fig. 2, we show the application of a model trained to predict the positions of Turow plumes. It was trained on pairs of fields in the regions of {Boxberg, Berlin, Lippendorf, Patnow, Jänschwalde, Dolna Odra, Schwarze Pumpe, Opole}. The first



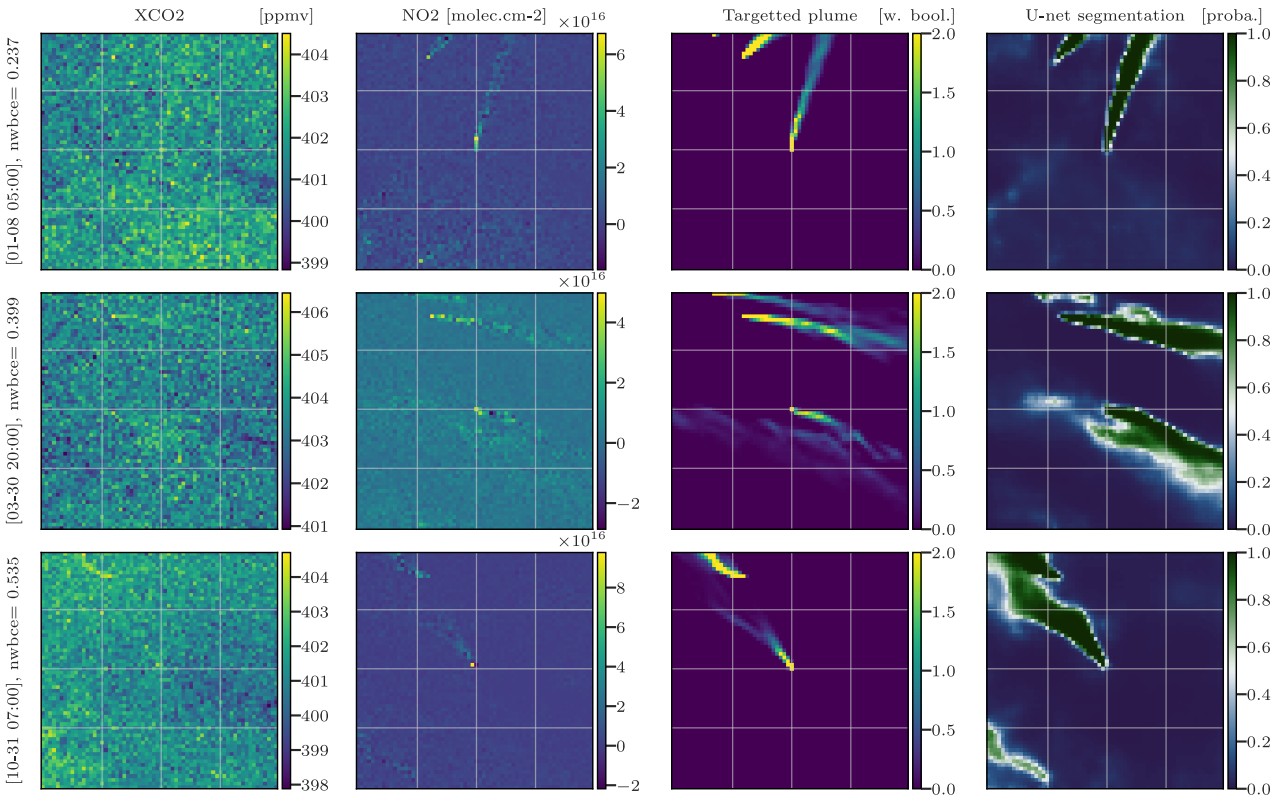

**Figure 2.** Examples of the U-net CNN model application on images centred at Turow. The first, second, third and fourth columns represent the $XCO_2$ images, $NO_2$ images, weighted Boolean plumes and U-net predictions as probability maps, respectively. All times are in UTC.

and second columns of the figure show the $XCO_2$ and $NO_2$ fields, inputs to the CNN. The third column shows the target plume as a reference point, while the fourth column shows the output of the CNN.

## 4 Deep learning method the for inversion of $XCO_2$

### 4.1 Inversion based on supervised learning

The inverse problem addressed here is the estimation of the $CO_2$ emissions accountable for the target plume observed in a given $XCO_2$ field image. To do so, a CNN is used, processing as input a given $XCO_2$ field and other additional fields and resulting in a scalar output representing the emission rate of $CO_2$ in $MtCO_2.yr^{-1}$ at the hour corresponding to the image. This flux unique scalar representation choice was made for the sake of simplification, as the quality of the results is minimally impacted by the choice between targeting average, instantaneous fluxes, or a vector of instantaneous fluxes over the last $N$ hours. This is due to the relatively slow hourly variation in the $CO_2$ emission rate of the PP. All future flux quantities are expressed in $Mt.yr^{-1}$.



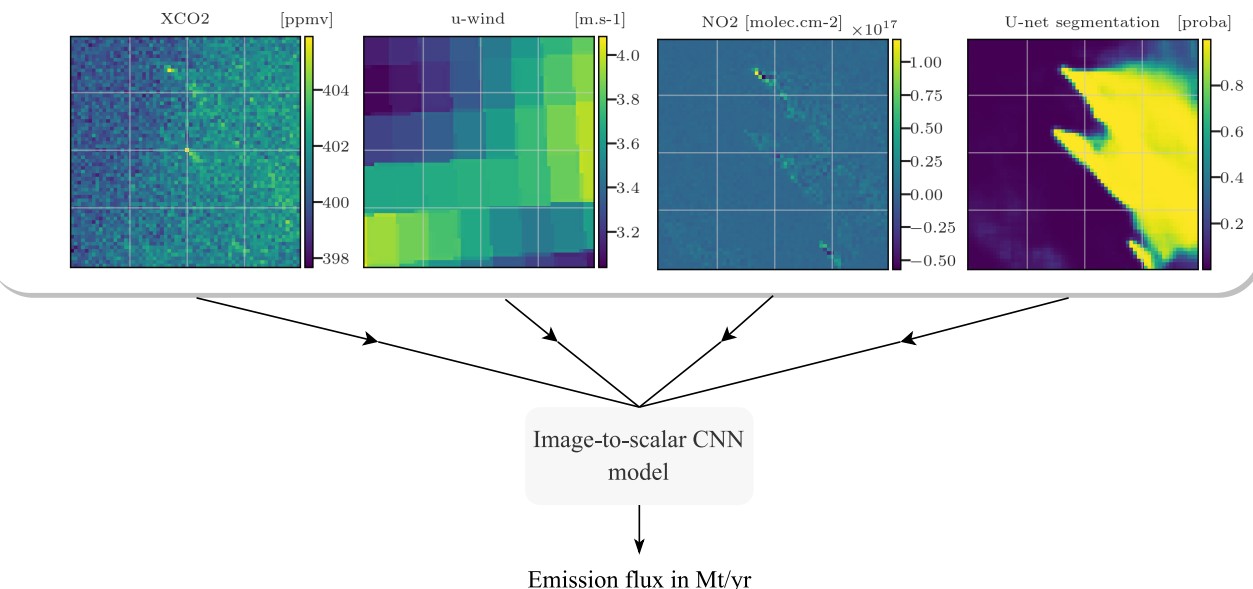

**Figure 3.** The XCO$_2$ fields, ancillary data, and emissions (of the central plume) are used as inputs by a CNN that learns to estimate the emissions associated to the central plume concealed under the background.

The image-to-scalar, or image regression, problem is depicted in Fig. 3. The CNN is trained using pairs of XCO$_2$ fields and associated emission fluxes ranging from 3 to $53\,\mathrm{Mt.yr}^{-1}$ across various times and targets.

## 4.2 CNN model and training parametrisation

In this section, we present and discuss the architecture of the model, the hyperparameters, and the learning methodology for the inversion task. In particular, the model is built from preprocessing layers and a core model. The preprocessing layers are used to augment/transform, construct, add noise, and normalise the input data before feeding it into the core model. The core model is designed to extract features from these transformed input data.

### 4.2.1 Description of the preprocessing layers

The preprocessing layers consist of a six steps sequence, as presented in Fig. 4. The purpose of steps 1-2-3 is to extend the initial database, thereby enhancing the model's ability to generalise to unseen data. A pair of input/output data for training is constituted in this way:

1. a target CO$_2$ plume corresponding to a PP at a time t is chosen randomly. A background must be constructed to form the XCO$_2$ image with this plume. This background is chosen randomly and therefore does not necessarily correspond (geographically and temporally) to the chosen target plume. It may correspond to another PP as well as to another time. Furthermore, in SMARTCARB, the background is partitioned into multiple segments. In our case, the background is





constructed from two randomly and independently drawn fields: a field containing the major part of the fluxes including
the biogenic fluxes and a field containing a part of the anthropogenic fluxes of the SMARTCARB domain. Finally,
potential additional fields (wind, $NO_2$, segmentations) corresponding geographically and temporally to the chosen $CO_2$
plume are selected;

2. then,

   – the target $XCO_2$ plume is multiplied by a random uniform scaling factor $p \sim U(0.25, 2)$. The corresponding true
     emissions are also multiplied: $y_{\text{truth}}^{\text{scaled plume}} = p \times y_{\text{truth}}^{\text{plume}}$;

   – a random number $b \sim U(-3.5, 3.5)$ (in ppmv) is added uniformly to the main background. Extrema of this uniform
     distribution are chosen as approximately the standard deviation of an average $XCO_2$ background;

   – a random uniform scaling factor $a \sim U(0.33, 3)$ is applied to multiply the field containing a part of the alternate
     anthropogenic fluxes.

3. the $XCO_2$ field is built as the sum of the target plume, background, and alternate anthropogenic fluxes field components;

4. a Gaussian noise matrix of shape $64 \times 64$ (equal to the image shape) $\mathbf{G}_{\text{Nx} \times \text{Ny}} \sim N(0, 0.7)$ (in ppmv) is added to the
   $XCO_2$ field to simulate the satellite observational noise;

5. the noisy $XCO_2$ field is concatenated with the additional fields. If added, the $NO_2$ field is noised beforehand;

6. standardisation is performed independently on each channel (each physical field) of the concatenated input data.

These steps are carried out exclusively during the model training phase. The different operations in this process are performed
to create a more robust and diverse training dataset. To ensure an accurate assessment of the performance of the trained model,
the test dataset used for the evaluation consists only of pre-constructed, physically consistent simulated data. Specifically, no
scaling factors are applied and the $XCO_2$ fields used for testing are always constructed from geographically and temporally
consistent plume and background components.

**4.2.2 Description of the core model**

The chosen core CNN model, described in Fig. 5 is designed for image regression. It was chosen by comparing its performance
with state-of-the-art models such as EfficientNet or Squeezenet (Tan and Le, 2020). These two models are deep neural networks
designed primarily for image classification tasks. They incorporate modified versions of CNN, including features such as
residual connections and depth-separable convolutions, in order to improve efficiency, speed and ease of implementation. As
their initial implementation tended to overfit, we considered a smaller version with a reduced number of neurons in each layer.
But even after tuning, the simpler model depicted below outperformed these more advanced models.

The chosen model takes 4 to 5 images of $64 \times 64$ pixels as input. It is constructed as a succession of convolutional, max
pooling, batch normalisation, and dropout layers where:





| Plume for chosen PP at time t | Background for random PP and time | Alt. anthro. fluxes for random PP and time |
|---|---|---|

$\times p \sim U(0.25, 2)$   $+b \sim U(-3.5, 3.5)$   $\times a \sim U(0.33, 3)$

$+$

$+\mathbf{G}_{\mathrm{Nx \times Ny}} \sim N(0, 0.7)$

**XCO2 field**

Concatenation ← Winds and additional fields corresponding to chosen plume

Normalisation

**Core CNN model processing**

**Backpropagation**

$$\mathrm{Loss} = \frac{pred - p \times y_{\mathrm{truth}}}{p \times y_{\mathrm{truth}}}$$

**Figure 4.** Description of the preprocessing layers as a sequence of six steps: 1) random choice of the XCO₂ field components, 2) scaling transformation of the components, 3) sum of the components, 4) satellite noise simulation, 5) concatenation with the additional data, and 6) normalisation.





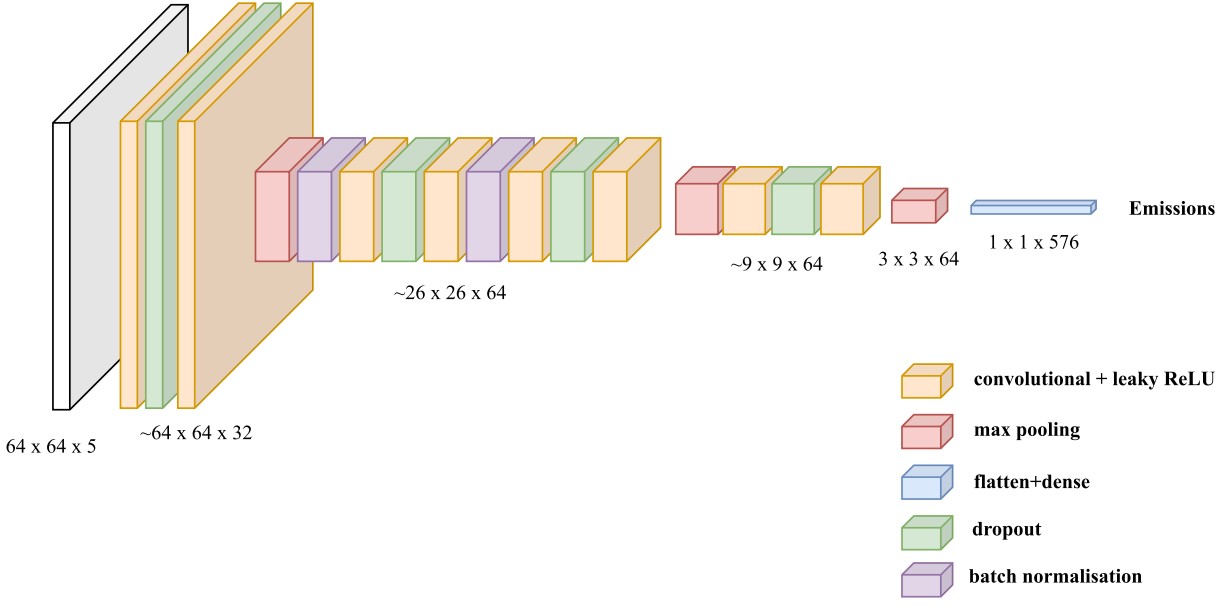

**Figure 5.** Description of the core CNN model, designed to extract features from the input data. The resolution given for each sequence of layers is approximate, as convolutional layers also slightly reduce the resolution.

- convolutional layers aim to identify and extract relevant features by applying a set of learnable filters to the previous
feature map. The 2D convolutional operations are applied with a filter size of 32 and a kernel size of $3\times3$;

- max pooling layers play a key role in reducing the resolution of feature maps while retaining the essential information. This reduces the computational complexity of the network and leads to the extraction of more complex features;

- batch normalisation layers are used to improve the stability of the network and speed up its learning process;

- dropout layers randomly exclude a certain percentage of the neurons in the previous layer at each iteration to reduce
overfitting. They are only activated during the training phase.

The output of the core model is flattened and fed into a fully connected terminal dense layer with a single output unit, which is activated by a leaking rectified linear unit (ReLU). In total, $\sim 186,000$ parameters are trained by the CNN.

### 4.2.3 Training parametrisation

The training hyperparameters, such as the optimiser, learning rate and batch size, have been determined through a combination
of experimental investigation and adherence to standard practice. In accordance with customary practices, Adam's optimiser was employed with a fixed learning rate of $10^{-3}$ and a dropout rate of $0.2$ was applied. The batch size, or number of samples before the model updates its weights, was set to 32. After analysis, to ensure model convergence, the epoch count was set to 500, which yields a total training time of approximately four hours using an Nvidia Quadro RTX 5000 16GB GPU. Furthermore, the



default choice of loss function employed is the Mean Absolute Percentage Error (MAPE), which emphasises proportionate deviations between emissions predictions and ground truth. This loss function is equivalent to a relative error between the truth and the predictions. Although, it should be noted that the Mean Absolute Error (MAE) was utilised to train roughly half of the ensemble models (detailed in the subsequent section 4.2.4) in the Lippendorf case (5.1). Both metrics generally yielded similar performances.

### 4.2.4   Model ensembling

In many cases, deep learning models suffer from high levels of volatility due to their reliance on random initialisations of parameters and hyperparameter optimisation algorithms. To overcome these limitations and increase the stability of our predictions, we employed the technique of model ensembling. Specifically, we trained several instances of the same model architecture and configured parameters, allowing us to generate diverse predictions that were subsequently averaged into a single estimate. To ensure sufficient convergence of these estimates, at least five individual models were used for each considered configuration. Through these steps, we significantly reduced the variability observed across multiple models runs, thereby improving the accuracy and confidence in our system.

### 4.3   Geographical separation between the training and test datasets

We consider a strict geographical separation between the PPs of the training/validation and those of the test dataset. For example, to train a model to predict emissions from Boxberg plumes, we consider a training dataset consisting of images centred at all other available PPs except Boxberg. In this way, the model predicting Boxberg emissions never had access to the Boxberg emissions inventory.

During the training phase, to avoid overfitting or underfitting, a separate validation dataset is used to follow the model's performance on new data. In our case, the validation dataset is chosen as unseen data from the same geographical area as the training dataset.

Three distinct models are created and trained to predict emissions from three specific PPs. The target PPs are Lippendorf, Boxberg and Turow and are selected to obtain:

- a diverse range of emission rates: between 7 and $24\,\mathrm{Mt.yr}^{-1}$, with a mean of $15\,\mathrm{Mt.yr}^{-1}$ and a std of $3\,\mathrm{Mt.yr}^{-1}$ for Lippendorf, between 4 and $13\,\mathrm{Mt.yr}^{-1}$, with a mean of $9\,\mathrm{Mt.yr}^{-1}$ and a std of $2\,\mathrm{Mt.yr}^{-1}$ for Turow, and between 9 to $30\,\mathrm{Mt.yr}^{-1}$ with a mean of $19\,\mathrm{Mt.yr}^{-1}$ and a std of $4.\,\mathrm{Mt.yr}^{-1}$ for Boxberg;

- the potential presence of multiple plumes in the same image: the images centred at Boxberg also include plumes from Jänschwalde, Schwarze Pumpe, and Turow which sometimes overlap.

### 4.4   Alternative comparison method

In order to assess the accuracy of our inversion method, we compare its predictions with those obtained from the cross-sectional fluxes method. This method consists in





– detecting the plume and extracting it from the background. $NO_2$ fields are used to help in the detection;

– dividing the plume into a series of horizontal slices of known areas and heights;

– estimating the line densities of $CO_2$ by fitting Gaussian curves to the $CO_2$ concentrations within each slice;

– inferring the $CO_2$ fluxes as the product of the line densities and the wind speed at the sources;

– deriving the total emission rate by multiplying the flux estimation and the area of each slice, and then averaging all
downstream fluxes.

The CSF method is limited by the need for accurate estimation of the effective wind (Kuhlmann et al., 2020). Two separate wind estimates have been considered: the first derived from an average of the 37 lower levels of ERA 5 data, and the second corresponding to the wind at 100 meters. The first estimate was chosen because of its superior performance. We implement the method with the Python package for data-driven emission quantification (ddeq[1]).

## 5 Application: inversion of three power plants and model interpretation

In this section, we study the performance of a trained CNN under the conditions exposed in section 4. Specifically, we analyse the potential of CNNs in the inversion of three different PPs:

– firstly, the model is evaluated on the Lippendorf PP in section 5.1. The training dataset is composed of {Berlin, Jän-schwalde, Schwarze Pumpe, Boxberg, Turow, Patnow, Opole, Dolna Odra}. Furthermore, we investigate how the assim-
ilation of $NO_2$ and segmentation fields affects the CNNs;

– secondly, in section 5.2, the model is investigated on the Turow PP, and is trained on {Berlin, Jänschwalde, Schwarze Pumpe, Boxberg, Patnow, Lippendorf, Opole, Dolna Odra};

– finally, the model is evaluated on the Boxberg PP in section 5.3. The training dataset includes {Berlin, Jänschwalde, Schwarze Pumpe, Turow, Patnow, Lippendorf, Opole, Dolna Odra}. Since issues of overfitting arise in certain configu-
rations, discussions and partial solutions to these issues are proposed;

In each configuration, the training, validation, and test datasets involve 25152, 4608, and 6289 images, respectively.

### 5.1    Inversion of Lippendorf plumes

### 5.1.1    Performance evaluation

We study the performance of three models trained with different sets of inputs to predict the emissions of the Lippendorf PP. Its
typical emissions fall between those of low-emission PPs (such as Dolna Odra or Turow) and those of high-emission (Boxberg,

---

[1]https://gitlab.com/empa503/remote-sensing/ddeq



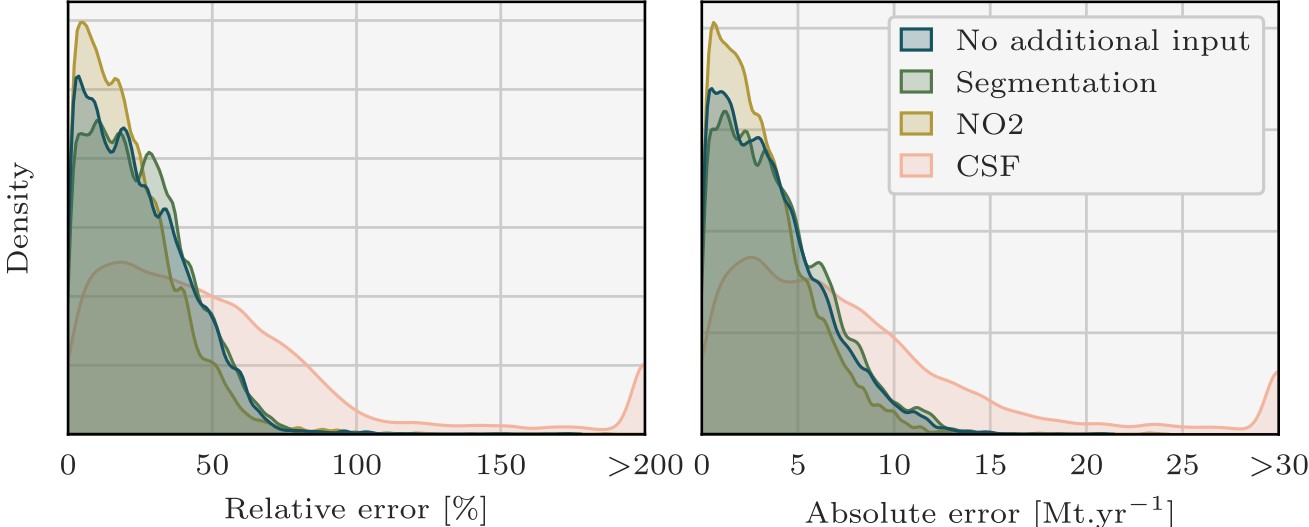

**Figure 6.** Density plots of the relative and absolute error between the predicted and the true Lippendorf emissions. Four sets of predictions are considered, corresponding to the three CNN models with three different ensembles of inputs and the CSF method. Each CNN model is trained with the $XCO_2$ field and the winds as inputs. Two of the models additionally assimilate the $NO_2$ field or the predictions of the segmentation model. Predictions with relative errors greater than $200\,\%$ or absolute errors greater than $30\,\mathrm{Mt.yr}^{-1}$ were set to 200 or 30 to increase visibility. $3\,\%$ of the CSF method predictions are missing. Those predictions correspond to Lippendorf plumes superimposed on other plumes, where the CSF method can not be applied.

Jänschwalde). All three models are trained on a dataset composed of {Berlin, Jänschwalde, Schwarze Pumpe, Boxberg, Turow, Patnow, Opole, Dolna Odra}. The input data for the first model are the $XCO_2$ field and the wind fields $u$ and $v$. The second and third models use the same three base fields and the output of the segmentation model, or the $NO_2$ field, respectively, as the fourth input.

Kernel Density Estimation (KDE) plots are drawn in Fig. 6 comparing with relative and absolute metrics the true emission rates and the predictions of four approaches. The three first ensembles of predictions consist of those of the trained CNNs, and the fourth corresponds to the CSF method application. The main statistics corresponding to these KDE plots are summarised in table 2.

The utilisation of the CNN approach yields remarkably accurate predictions compared to the CSF method. The performance
of all three CNN models shows a median relative error of approximately $20\,\%$ and a median absolute error of around $3\,\mathrm{Mt.yr}^{-1}$ (where the average emissions for Lippendorf are $15.2\,\mathrm{Mt.yr}^{-1}$). In comparison, the CSF method exhibits a higher median relative error performance of around $40\,\%$, and the absolute error performance is approximately double, at $6\,\mathrm{Mt.yr}^{-1}$. These results align with the findings reported in (Kuhlmann et al., 2021) and (Hakkarainen et al., 2023). Other methods, such as the





| | Relative error [%] | | | Absolute error [Mt.yr$^{-1}$] | | |
|---|---|---|---|---|---|---|
| | 25% | Median | 75% | 25% | Median | 75% |
| CNN with no additional input | 9.8 | 21.3 | 35.6 | 1.4 | 3.1 | 5.2 |
| CNN with segmentation | 11.4 | 23.3 | 36.5 | 1.6 | 3.4 | 5.6 |
| CNN with NO$_2$ | 8.6 | 18.1 | 30.3 | 1.3 | 2.7 | 4.5 |
| CSF | 21.0 | 42.8 | 70.1 | 3.1 | 6.3 | 10.5 |

**Table 2.** Relative and absolute error statistics between predicted and true Lippendorf emissions for three CNN models (assimilating three different sets of inputs) and the CSF method.

light cross sectional fluxes technique, might yield predictions with relative errors reduced by 5 to 10% (Hakkarainen et al., 2023).

A notable constraint of the CSF method is that it hinges on the estimation of the effective wind speed inside the plume. Under ideal circumstances, the effective wind speed should be estimated using the wind profile, weighted by the CO$_2$ concentration profile, but in practical applications, estimating this profile presents a substantial challenge. In a separate experiment, the CSF method was utilised on a reduced dataset (compared with that used to study the CNN results). Among other changes, this dataset considers only the CO$_2$M overpass, or excludes the plume overlaps processed by the CNN. In this experiment, the exact emission profile used to simulate the COSMO-GHG fields of SMARTCARB was used to compute the effective wind. This methodology resulted in a median absolute error around 4 Mt.yr$^{-1}$, thereby indicating that the effective wind estimation significantly contributes to the errors associated with the CSF method. The CNN results are reliable, as the majority of errors are concentrated below 10 Mt.yr$^{-1}$ or 50%, with very few exceeding 100%. This indicates that the models provide trustworthy estimates, with a relatively small margin of error.

### 5.1.2 Study of the addition of the segmentation and NO$_2$ fields

Surprisingly, the addition of the segmentation results in the inputs does not improve the inversion process. One possible reason for this lack of improvement is that the Lippendorf segmentation fields are inadequate. The segmentation model assimilates NO$_2$ fields to segment the CO$_2$ fields and the presence of multiple alternative NO$_2$ plumes in the NO$_2$ fields hinders the segmentation model's ability to accurately delineate the contour of the Lippendorf CO$_2$ plume. However, the incorporation of NO$_2$ field as inputs slightly increase the quality of the predictions made by the new CNN model. One hypothesis to explain the error discrepancy between the model utilising NO$_2$ and the one using segmentation fields is due to the segmentation model not capturing NO$_2$ or CO$_2$ plume amplitude variations. Precisely, the segmentation model does not discriminate between plume pixels with high amplitude and those with low amplitude. Consequently, if the NO$_2$ images are of poor quality due to the presence of numerous alternative NO$_2$ plumes, the segmentation model will struggle to distinguish the main plume accurately.

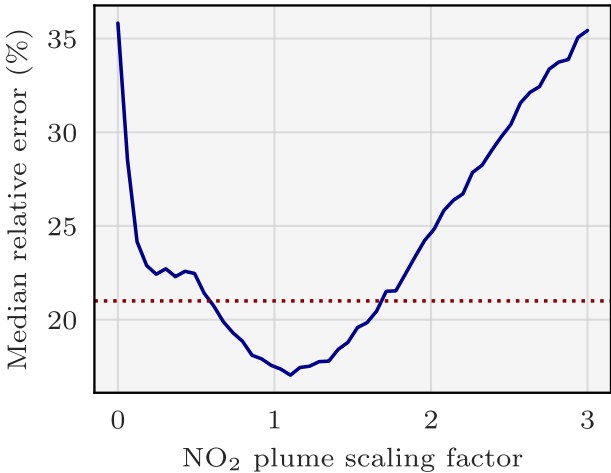

**Figure 7.** Effect of scaling the $NO_2$ plume of Lippendorf inputs on the performance of the CNN model. The x-axis corresponds to the scaling of the $NO_2$ plume: 1 corresponds to the original plume, 0 to no plume. The y-axis corresponds to the median relative error of the CNN model evaluated for the given scaling of the $NO_2$ plumes. The CNN model is the same for each scaling (each dot of the x-axis) and corresponds to the CNN model having obtained the best relative error score. The red dotted line approximately corresponds to the median relative error of a model not learning with $NO_2$ fields.

Taking into account these segmentation fields, which do not distinguish the background plumes in the test dataset, results in a degradation of the predictions.

Regarding the benefits brought by the integration of $NO_2$ fields, their essential contribution is their facilitation of the plume segmentation, rather than a direct enablement of inversion based on the $NO_2$ levels alone. Indeed, as stated in section 4.2.1, the
$NO_2$ plumes are not scaled like $CO_2$ plumes. To further investigate this, it is possible to verify this hypothesis by modifying the scaling of the plume in the $NO_2$ fields. In Fig. 7 we draw the relative error of the model when taking as inputs scaled $NO_2$ fields by a constant factor in $[0, 3]$. The CNN model exhibits its highest performance within the scaling range of 1.0 to 1.3. As expected, performances gradually decrease for scalings below 1 and above 1.3. However, intriguingly, the model still achieves remarkably satisfactory scores for scalings ranging from 0.5 to 1.75. Notably, these scores outperform those of the
ensemble of CNN models without $NO_2$ field as input. If the model were utilising the amplitude of the $NO_2$ field as a predictor, deeply inaccurate results could be expected for scalings of 0.5 or 1.5. However, the graph contradicts this assumption, strongly suggesting that the amplitude of the $NO_2$ field is not employed as a predictor by the model. Instead, it is likely that only the contour of the $NO_2$ plume and the ratios between different parts of the plume serve as predictors. In essence, the model's reliance on the $NO_2$ field for predictions appears to be based on the contour of the plume and the relative proportions of its
various components, rather than on the absolute amplitude of the $NO_2$ field values. Finally, the model's inability to accurately



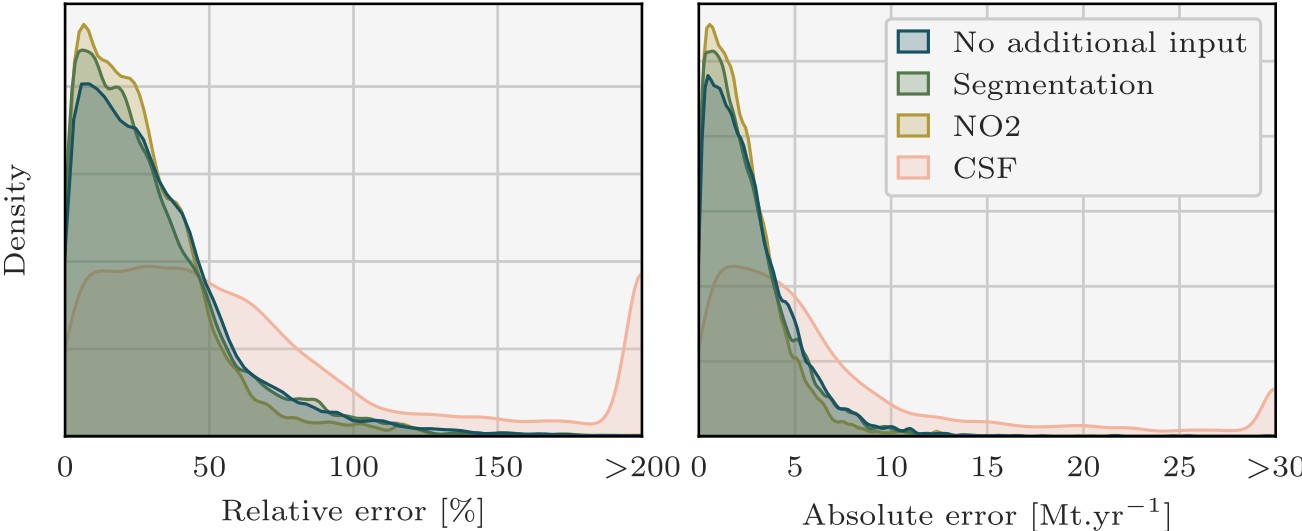

**Figure 8.** Density plots of the relative and absolute error between the predicted and the true Turow emissions. Four sets of predictions are considered, corresponding to the three CNN models with three different ensembles of inputs and the CSF method. Each CNN model is trained with the $XCO_2$ field and the winds as inputs. Two of the models additionally assimilate the $NO_2$ field or the predictions of the segmentation model. Predictions with relative errors greater than $200\%$ or absolute errors greater than $30\,\mathrm{Mt.yr^{-1}}$ were set to 200 or 30 to increase visibility.

estimate emissions for scaling factors exceeding 2 can be attributed to the unprecedented and extreme values that $NO_2$ plumes can reach, which lie beyond the range of what the model has been trained on.

## 5.2 Inversion of Turow plumes

### 5.2.1 Performance evaluation

In this section, we study the performance of three models with input similar to those of section 5.1.1, with this time a focus on a low-emission PP: Turow. Turow emissions range between $4$ and $14\,\mathrm{Mt.yr^{-1}}$, similar to PPs like Dolna Odra or Opole. This implies that most Turow $CO_2$ plumes are hidden under the background. All three models considered are trained on a dataset consisting of {Berlin, Jänschwalde, Schwarze Pumpe, Boxberg, Lippendorf, Patnow, Opole, Dolna Odra}.

In a similar manner to section 5.1.1, KDE plots are presented in Fig. 8 to compare the true emission rates with the predictions

of the four different approaches using relative and absolute metrics and corresponding statistics are summarised in table 3.

The CNN approach produces highly accurate predictions for these low SNR plumes, exhibiting similar performance to the results obtained in the Lippendorf case. The three models yield a median relative error performance of approximately $25\%$





| | Relative error [%] | | | Absolute error [Mt.yr$^{-1}$] | | |
|---|---|---|---|---|---|---|
| | 25% | Median | 75% | 25% | Median | 75% |
| CNN with no additional input | 12.3 | 25.9 | 43.2 | 1.0 | 2.2 | 3.8 |
| CNN with segmentation | 11.0 | 23.7 | 41.3 | 0.9 | 2.0 | 3.5 |
| CNN with NO$_2$ | 10.9 | 22.9 | 38.0 | 0.9 | 1.9 | 3.2 |
| CSF | 26.3 | 52.1 | 92.0 | 2.2 | 4.5 | 8.1 |

**Table 3.** Relative and absolute error statistics between predicted and true Turow emissions for three CNN models (assimilating three different sets of inputs) and the CSF method.

and a median absolute error performance of around $2\,\mathrm{Mt.yr}^{-1}$. The results can be considered reliable: $75\%$ of the results fall below a threshold of $4\,\mathrm{Mt.yr}^{-1}$. By contrast, the CSF method exhibits a median relative error performance of approximately 320 $50-55\%$ and an absolute error performance more than two times larger.

The inclusion of segmentation or NO$_2$ fields has a noticeable, albeit not significant, impact on the model's performance, resulting in an improvement on the order of a tenth of a $\mathrm{Mt.yr}^{-1}$. Notably, the addition of the NO$_2$ field appears to have a slightly greater impact compared to the inclusion of segmentation fields. This implies that when applying the model that assimilates the segmented XCO$_2$ fields (with the assistance of the NO$_2$ fields), there is a potential loss of information on the 325 NO$_2$ fields compared to using the model directly assimilating the NO$_2$ fields. The observed phenomenon could be a result of potential overfitting when employing the segmentation model, as it is trained on the same dataset as the inversion model. Consequently, the segmentation predictions on the training dataset are typically superior to those on the test dataset. This discrepancy can lead to an over-reliance on the segmentation fields, subsequently causing overfitting.

### 5.3 Inversion of Boxberg plumes

#### 5.3.1 Performance evaluation

The last studied PP is Boxberg which is characterised by high emissions and the presence of other PPs in the vicinity, which entails the presence of other high SNR plumes. We study the performance of three models, all trained on a dataset consisting of {Berlin, Jänschwalde, Schwarze Pumpe, Lippendorf, Turow, Patnow, Opole, Dolna Odra}. The three models differ in their inputs, mirroring the approach taken in sections 5.1 and 5.2. Figure 9 and table 4 present a comparison between the true 335 emission rates and the corresponding predictions.

In contrast to the inversion results obtained for Lippendorf and Turow PPs, there is a significant variation in performance among the three CNN models. A CNN only trained with XCO$_2$, $u$ and $v$ winds demonstrates strong performance, comparable to those of the models estimating Lippendorf and Turow emissions in sections 5.1 and 5.2. The median relative error performance



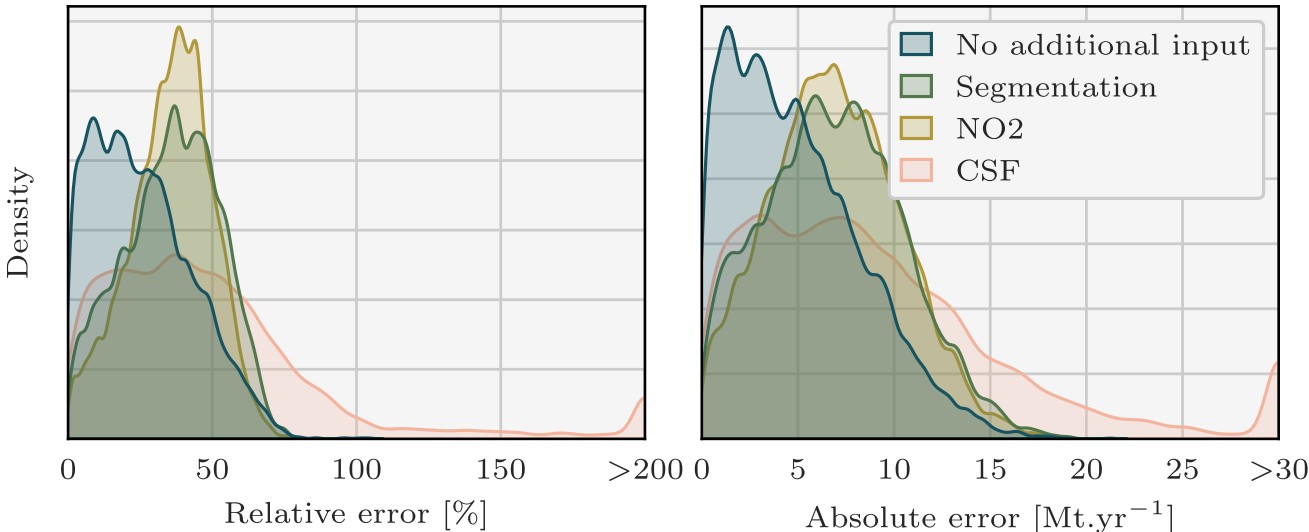

**Figure 9.** Density plots of the relative and absolute error between the predicted and the true Boxberg emissions. Four sets of predictions are considered, corresponding to the three CNN models with three different ensembles of inputs and the CSF method. Each CNN model is trained with the $XCO_2$ field and the winds as inputs on a dataset composed of {Berlin, Jänschwalde, Schwarze Pumpe, Lippendorf, Turow, Patnow, Opole, Dolna Odra}. Two of the models additionally assimilate the $NO_2$ field or the predictions of the segmentation model. Predictions with relative errors greater than $200\,\%$ or absolute errors greater than $30\,\mathrm{Mt.yr^{-1}}$ were set to 200 or 30 to increase visibility. $15\,\%$ of the CSF method predictions are missing. Those predictions correspond to Boxberg plumes superimposed on other plumes, where the CSF method cannot be applied.

| | Relative error [%] | | | Absolute error [$\mathrm{Mt.yr^{-1}}$] | | |
|---|---|---|---|---|---|---|
| | 25% | Median | 75% | 25% | Median | 75% |
| CNN with no additional input | 11.7 | 23.5 | 37.2 | 2.1 | 4.4 | 7.1 |
| CNN with segmentation | 24.2 | 36.9 | 48.2 | 4.2 | 6.8 | 9.4 |
| CNN with $NO_2$ | 26.4 | 36.9 | 45.4 | 4.5 | 6.8 | 9.3 |
| CSF | 21.7 | 41.5 | 63.5 | 3.9 | 7.7 | 12.3 |

**Table 4.** Relative and absolute error statistics between predicted and true Boxberg emissions for three CNN models (assimilating three different sets of inputs) and the CSF method.

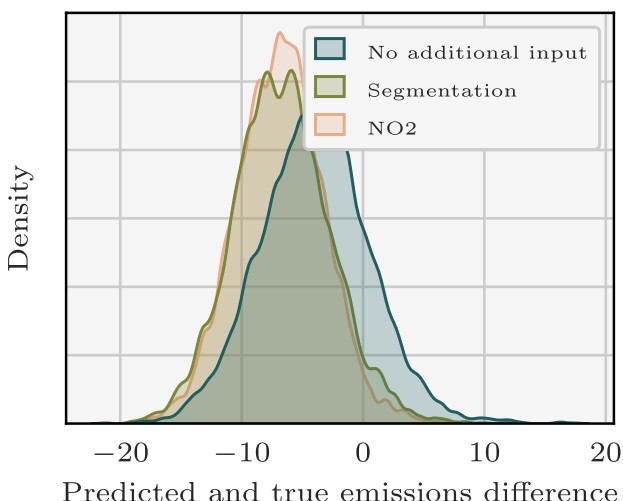

**Figure 10.** Residuals density between the true emissions of Boxberg PP and the predictions of the CNNs.

is approximately $20 - 25\%$, and the median absolute error performance is around $4\,\mathrm{Mt.yr}^{-1}$. By contrast, the CSF method
exhibits a median relative error performance of around $40\%$ and an absolute error performance close to $8\,\mathrm{Mt.yr}^{-1}$. But both
models with segmentation or $NO_2$ fields show a significant decline in performance, which contradicts our expectations. This
phenomenon can be attributed to overfitting and is examined in detail in the subsequent sub-section.

### 5.3.2 Overfitting investigation

To understand the reasons behind the deviations between predictions and actual values, we conducted an analysis of the resid-
uals in Fig. 10. This examination, which focuses on the disparities between predicted and true emissions, reveals a substantial
underestimation of the actual Boxberg emissions by the model incorporating $NO_2$ or segmentation fields.

This observation suggests an overfitting issue, since the majority of PPs used to train the model exhibit lower emission
rates compared to Boxberg (6 out of 7: Schwarze Pumpe, Lippendorf, Turow, Patnow, Opole, and Dolna Odra). The low
relative error observed on the training dataset of the models with $NO_2$ or segmentation fields, as depicted in Fig. 11, further
substantiates concerns regarding overfitting. An overfitting model tends to learn features that are overly tailored to the specifics
of the training dataset. Consequently, when presented with images from the test dataset with slightly different features, the
model struggles to generate accurate predictions.

Our hypothesis is that the model specifically overfits the low emissions PPs, which can be attributed to the information
gained through the use of $NO_2$ or segmentation fields. When provided with new images, the model fails to recognise the new
features and consequently yields predictions that align more closely with the training dataset, which predominantly consists of



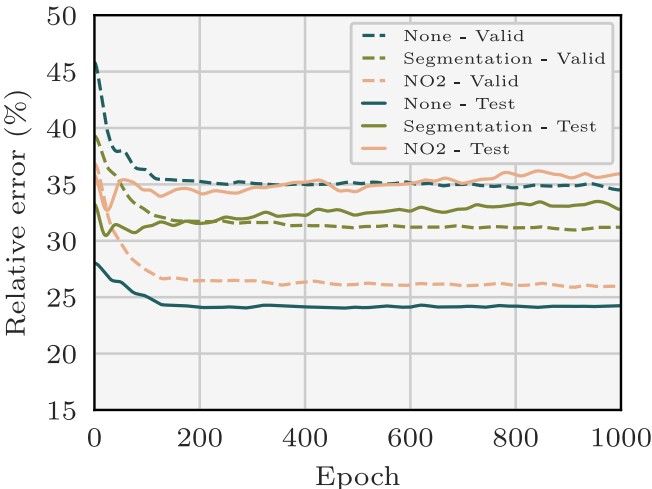

**Figure 11.** Evolution during training of the validation and test relative errors between the true emissions of Boxberg and the predictions of the CNNs. Specifically, for each epoch, the models being trained are employed to predict the emissions corresponding to the validation and test fields.

emissions from the low emissions PPs. The failure to recognise new features is due to the incorporation of $NO_2$ field as inputs. The model's ability to learn highly specific features is limited when no additional input is provided. Conversely, when the model incorporates the $NO_2$ field, it gains access to more information and can acquire more intricate features. Consequently, the model's capacity to generalise worsens in the latter case.

It is worth noting that some CNN models that we trained incorporating $NO_2$ or segmentation fields do not exhibit overfitting and provide highly accurate predictions. However, these models do not perform better on the validation dataset, rendering them unsuitable for selection.

Next, we examine the performance of a new ensemble of models trained on a more balanced dataset, achieved by removing three out of the five low-emissions PP from the original dataset. The new dataset is composed of {Berlin, Jänschwalde, Lip-

pendorf, Turow, Opole}, i.e. of two medium or high emissions PPs, two low emissions PPs and Berlin. Furthermore, the choice of the factor scaling the plumes during training (see section 4.2.1) varies depending on the ensemble member considered (see section 4.2.4). Specifically, the uniform distribution is defined with a minimum scaling factor of $0.25$ or $0.5$ and a maximum scaling factor of $2$ or $3$. Once more, we examine the outcomes for three models obtained from ensembling, as depicted in Fig. 12, and summarised in table 5.

The three CNN models demonstrate very good performance, although the inclusion of $NO_2$ or segmentation fields still leads to a degradation in results, albeit to a lesser extent than in section 5.3.1. On the one hand, the median relative error of the CNN model trained on $NO_2$ and wind fields stands at approximately $20\%$, while the median absolute error remains below

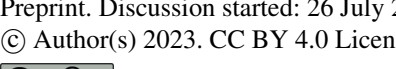



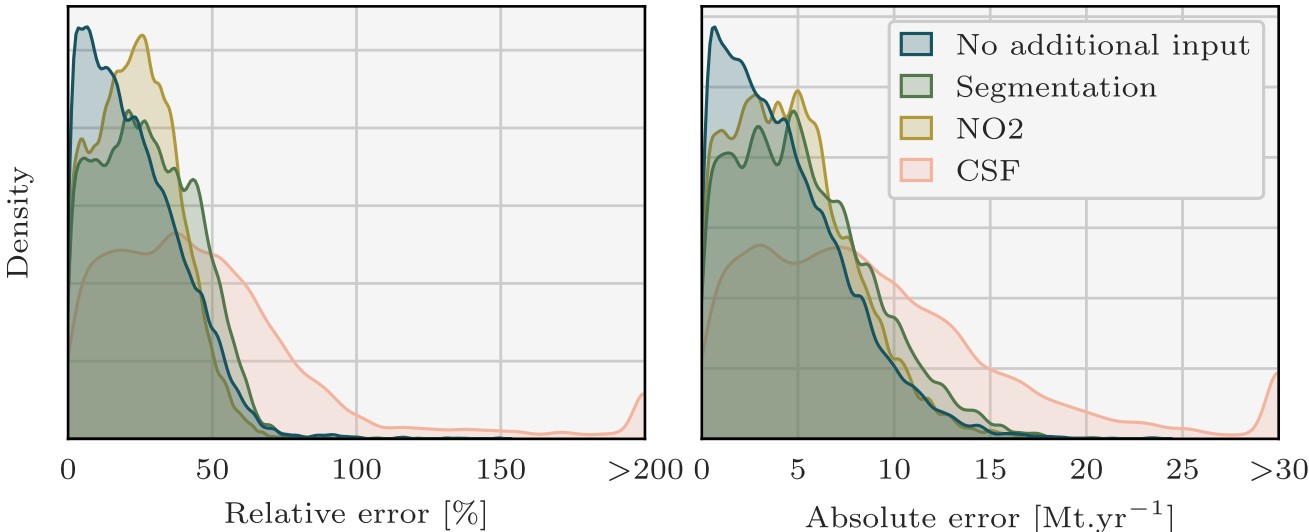

**Figure 12.** Density plots of the relative and absolute error between the predicted and the true emissions. Four sets of predictions are considered, corresponding to the three CNN models with three different ensembles of inputs and the CSF method. Each CNN model is trained with the $XCO_2$ field and the winds as inputs on a dataset composed of {Berlin, Jänschwalde, Lippendorf, Turow, Opole}. Two of the models additionally assimilate the $NO_2$ field or the predictions of the segmentation model. Predictions with relative errors greater than 200% or absolute errors greater than $30\,\mathrm{Mt.yr}^{-1}$ were set to 200 or 30 to increase visibility. 15% of the CSF method predictions are missing. Those predictions correspond to Boxberg plumes superimposed on other plumes, where the CSF method cannot be applied. A modified dataset is used to avoid overfitting.

|  | Relative error [%] | | | Absolute error [$\mathrm{Mt.yr}^{-1}$] | | |
|---|---|---|---|---|---|---|
|  | 25% | Median | 75% | 25% | Median | 75% |
| CNN with no additional input | 9.5 | 20.4 | 33.8 | 1.8 | 3.7 | 6.3 |
| CNN with segmentation | 14.1 | 26.6 | 40.6 | 2.5 | 4.8 | 7.6 |
| CNN with $NO_2$ | 13.2 | 23.8 | 34.2 | 2.3 | 4.4 | 6.6 |
| CSF | 21.7 | 41.5 | 63.5 | 3.9 | 7.7 | 12.3 |

**Table 5.** Relative and absolute error statistics between predicted and true Boxberg emissions for three CNN models (assimilating three different sets of inputs) and the CSF method. A modified dataset is used to avoid overfitting.





$4 \, \mathrm{Mt.yr}^{-1}$. On the other hand, the degradation of the results when adding the $NO_2$ or segmentation fields still can be regarded as overfitting. The model learns features from $NO_2$ or segmentation fields that are not general enough to cover the case of

Boxberg. Furthermore, it fails to acquire compensatory generalisable features such as in the case of Turow, where the model probably gains information about the plume contour from the $NO_2$ field, which is not straightforwardly apparent in the Turow $XCO_2$ field.

## 5.4 Interpretation of the CNN inversion models

In the two following sections, we introduce and apply two methods to gain a deeper understanding of the behaviour and

decision-making processes of the CNNs discussed in this paper. These methods offer valuable insights into the significance of input features in the predictions made by the CNNs:

- the integrated-gradient method allows us to examine the importance of individual pixels across channels;

- the feature permutation method enables us to assess the importance of the channels, i.e. the fields used as inputs.

### 5.4.1 Gradient-based study of the pixels

Integrated Gradients is a gradient-based method for the interpretability of neural networks that enables the assessment of pixel importance in CNN prediction. It calculates the sensitivity of a model's predictions to input features (here pixels), assigning relevance scores to them. By analysing how changes in the input pixels affect the model's output, the method provides insights into the importance of each pixel in the decision-making process.

One unique aspect of this gradient-based method is that the scores it assigns are relative to a baseline. More precisely, these

scores are computed as integrated gradients along a linear interpolation from a blank image to the input.

In Fig. 13, we apply the Integrated Gradient approach to study four different models specific to various sources. The first and second models are built to invert the emissions from Lippendorf (see section 5.1) and Boxberg plumes (see section 5.3), respectively, whereas the two last models target Turow plumes (see section 5.2). The first three models only use the $XCO_2$ field and the winds as inputs, whereas the fourth model considers an additional input, the $NO_2$ field. To apply the Integrated

Gradient method, a random plume from the target PP is selected for each model. Columns of Fig. 13 represent the $XCO_2$ field, the corresponding $XCO_2$ plume and the Integrated Gradient between the model predictions and the inputs.

In order to simplify the analysis, we choose the model that exhibits the best performance on the test dataset, rather than using the ensemble of models. It is worth noting that such a model yields similar performances to the ensemble of models.

The Integrated Gradient technique reveals that the CNN model learns to estimate the emissions of a source based on the

pixels of the plume from this source. In the first row, we examine a plume from Lippendorf PP. The Integrated Gradient technique identifies the most important pixels, which correspond to the plume pixels. This indicates that if the pixels associated with the plume were to deviate, the estimated flux rate would be significantly affected. This demonstrates that the CNN model effectively makes inference from the crucial parts of the image. In the second row, we focus on an image centred on Boxberg. The gradients reveal that the model concentrates exclusively on the Boxberg plume in the centre, disregarding the other plumes





**Figure 13.** Evaluation of four CNN models using the Integrated Gradients method on four input sets. Columns 1, 2, and 3 represent the $XCO_2$ field, the corresponding $XCO_2$ plume and the Integrated Gradient between the model predictions and the inputs, in that order. The rows represent the four models and corresponding test fields.



| PP | Lippendorf | | | Boxberg | | | Turow | | |
|---|---|---|---|---|---|---|---|---|---|
| Feature | None | Seg. | NO$_2$ | None | Seg. | NO$_2$ | None | Seg. | NO$_2$ |
| XCO$_2$ | 18.3 | 27.6 | 32.7 | 19.6 | 29.5 | 34.5 | 12.6 | 9.8 | 22.7 |
| u-wind | 11.6 | 10.4 | 3.3 | 12.4 | 9.7 | 4.2 | 6.6 | 5.4 | 0.9 |
| v-wind | 4.9 | 3.7 | 1.8 | 5.3 | 4.9 | 3.0 | 3.6 | 1.7 | 0.6 |
| Fourth feature | n/a | 22.7 | 30.3 | n/a | 28.9 | 34.6 | n/a | 6.9 | 21.0 |

**Table 6.** Evaluation of the degradation in the average relative error of the model when the corresponding feature is permuted. Each column corresponds to a model, and each row corresponds to a permuted feature. For example, we estimate the emissions corresponding to Boxberg plumes with the CNN model trained with the XCO$_2$ field, the winds and no additional input. We then estimate the degradation of the predictions of this model on the test dataset with the u-wind fields permuted: the degradation in relative error is $12.4\%$. Several cells of importance of this table are coloured and discussed thereafter.

when inferring the emissions of Boxberg PP. In the third row of the figure, we examine a plume from Turow PP. Although the precision is lower compared to the Lippendorf or Boxberg cases, the pixels in the general direction of the Turow plume are the main ones used to estimate the emissions. In the fourth row, we display the gradients associated with the same Turow image but for a model trained with NO$_2$ fields as additional inputs. In this case, the model clearly identifies the pixels corresponding to the plume as critical, as indicated by the amplitude and contour of the gradients. This reinforces the hypothesis that the improved estimation of Turow emissions when the model is trained with NO$_2$ fields, can be attributed to the enhanced assessment of the plumes.

### 5.4.2 Feature permutation analysis

Feature permutation is a technique used to determine the importance of input channels used in a model. The principle is to i) permute the values of a feature (e.g. exchange the $u$ wind field corresponding to an XCO$_2$ image with another random $u$ wind field), and ii) use the model to predict emissions for the given input, which includes the XCO$_2$ field, other associated inputs and the random $u$ wind field. By comparing the performance of the model on the original dataset with the performance on the permuted dataset, we can measure the impact of each feature on the performance of the model. The more the permutation of a feature affects the performance of the model, the more important that feature is. In table 6, we present the outcomes of the permutation of the features for nine ensembles of models (for each PP and each ensemble of inputs considered). Each entry in the table represents the degradation in the average relative error of the model (associated with a specific column) when the corresponding feature is permuted (related to the respective row).

We can formulate hypotheses on several groups of cells in table 6:





– in the highlighted yellow instances, we verify the anticipated observation that the $XCO_2$ feature consistently holds (or shares) the highest importance among other features (here, in comparison to $u$ wind) or shares the highest level of importance;

– in the highlighted orange section, we emphasise a trend indicating that u-wind appears to have a greater impact on the inversion than v-wind. It might be due to the bigger variance of $u$ ($\sim 51\,\mathrm{m^2.s^{-2}}$) comparing to $v$ ($\sim 19\,\mathrm{m^2.s^{-2}}$): $u$ is the dominant wind in this situation;

– purple cells are examples of the $XCO_2$ field inputs having less importance for the CNN targeting Turow. The observed behaviour is due to the low SNR of Turow plumes. Due to inconsistent ancillary (winds, ...) data and Turow plume, the model tends to estimate emissions from points in the image other than the plume. These points are less distinct from the plume than in the scenario with the Boxberg plume. As a result, the relative error is lower;

– in the highlighted red section, we underline the increase of the $XCO_2$ feature importance when additional data are used. This observation is consistent with the overfitting tendency of the CNN models when trained with, for example, $NO_2$ field. When confronted with inconsistent data (non-corresponding $XCO_2$ and $NO_2$ fields), the acquired complex features of the model exhibit a total absence of correspondence with these inputs. As a consequence, the model predicts nonsensical emissions;

– the brown cells reveal that the wind systematically holds greater importance in the model without additional input, followed by a relatively diminished importance in the model with segmentation fields, and finally, it exhibits the least significance in the model with $NO_2$ fields. This suggests that the inclusion of $NO_2$ fields is more advantageous for inversion compared to segmentation fields. Furthermore, it implies that the winds are used to compensate for the absence or deficiency of additional data.

## 6 Discussions and limitations

The approach developed in this paper carries certain limitations: firstly, it focuses exclusively on European PPs, which questions its generalisability to PPs in other regions with different climatic conditions. Secondly, the study emphasises the importance of a balanced dataset, as highlighted by the 5.3.2 section, and the need to be able to identify and address potential overfitting issues. It should also be noted that the study did not specifically investigate outliers such as PPs with exceptionally high emission rates. Despite incorporating plume scaling strategies, the model may struggle to generalise to such outliers.

In terms of future research, several areas should be explored such as the challenge posed by clouds. In this respect, CNNs can be trained to ignore missing data caused by cloud cover and to make effective use of the available data. Another aspect to consider is the presence of noise in $CO_2M$ data. While Gaussian noise may not pose significant issues, if the satellite noise exhibits structured patterns, it would becomes crucial to develop robust noise modelling techniques to enable CNNs to accurately distinguish and remove such noise. Finally, in real-world applications, training on synthetic datasets and applying





the trained models to real datasets may encounter distributional differences. Strategies such as importance weighting, specific
data augmentation techniques, transfer learning, or active learning methods may be necessary to account for these differences
and ensure reliable performance.

## 7  Conclusions and perspectives

The future availability of $CO_2$ satellite imagery, through missions such as the Copernicus $CO_2$ Monitoring ($CO_2$M) mission,
heralds new oppurtunities and challenges for the evaluation of local $CO_2$ emissions. Emissions can certainly be retrieved from
$CO_2$ plumes of hotspots in the satellite images. But the emission estimation is hindered by two primary obstacles: low signal-
to-noise ratio plumes which cannot be extracted straightforwardly from the background, and uncertainties in the transport or
dispersion processes which hampers the assessment of the emissions from the plumes.

In this work, we assess the ability of Convolutional Neural Networks (CNNs) to invert a plume from satellite imagery using
simulated $XCO_2$, $NO_2$, and wind fields with similar characteristics to the future $CO_2$M images. The fields used to train and
evaluate the CNNs are based on the SMARTCARB dataset and possess the same resolution, satellite noise level and ancillary
data availability to the future $CO_2$M images. Each synthetic $XCO_2$ image encompasses the anthropogenic plume from at least
one power plant, a background arising from other biogenic and anthropogenic fluxes, and a random Gaussian noise intended
to simulate the errors inherent in satellite instruments. But the model evaluation was conducted using simulated data. This
approach does not account for all the challenges that real satellite images present, specifically issues related to cloud cover and
systematic error patterns due to surface reflectance and the aerosol dependency of retrievals.

Our source emission estimation model is an image-to-scalar CNN model, which infers from the full $XCO_2$ field a flux rate,
estimation of the anthropogenic emission corresponding to the plume. The first layers of the model consist of preprocessing
steps which transform the input data at training time, in particular by adding noise and by scaling the plume and emission rate.
The core model is a CNN composed of approximately 200,000 neurons divided into convolutional, max pooling, dropout and
batch normalisation layers.

We demonstrated that the design of a "universal" CNN, trained on a small power plant subset and highly accurate on all of
them, is possible. To do so, we evaluated the model's ability to generalise to unobserved images from another region. Explicitly,
three CNN models were tested on plumes from three sources: Boxberg, Lippendorf and Turow. The training dataset for each
CNN is restricted to a dataset consisting of all other power plants except their target. The CNNs are highly accurate in each
case and the addition of $NO_2$ fields often improves the results slightly. Precisely, the median relative errors for the CNN models
are on average close to 20–25 %. Moreover, the median absolute error is generally half that obtained with the CSF method: an
alternative and state of the art inversion approach.

By using interpretability tools, we demonstrate that the predictions made by the CNNs are grounded in the physically
meaningful components of the features. The Integrated Gradient method shows that the CNNs learn to predict the emissions
corresponding to a plume from the pixels making up the plume. The feature permutation technique highlighted several aspects
of the models, such as the expected high importance of the $XCO_2$ fields compared to the used ancillary data.



Future prospects of the CNN plume inversion method from satellite images encompass the challenges of clouds, cities and real satellite images. Concretely, the method should be able to handle missing data caused by clouds. Additionally, the CNN approach should incorporate the second important category of hotspots: cities. Finally, the method should be tested on real
satellite data once it becomes available.

*Data availability.* The datasets used in this paper are available on a compliant repository on https://zenodo.org/record/8096616 and originate from https://zenodo.org/record/4048228. The weights of the CNNs are available on https://zenodo.org/record/8095487. The algorithms are available on Zenodo (https://zenodo.org/record/8096760) and Github at https://github.com/cerea-daml/co2-images-inv-dl.

*Author contributions.* Joffrey Dumont Le Brazidec: Conceptualisation, Methodology, Software, Investigation, Formal analysis, Visualisa-
tion, Resources, Project administration, Writing - Original Draft; Pierre Vanderbecken: Investigation, Formal analysis, Writing - Review; Alban Farchi: Conceptualisation, Methodology, Project administration, Writing - Review; Marc Bocquet: Conceptualisation, Methodology, Project administration, Funding acquisition, Writing - Review; Grégoire Broquet: Conceptualisation, Writing - Review; Gerrit Kuhlmann: Resources, Writing - Review.

*Competing interests.* The authors declare that they have no conflict of interest.

*Acknowledgements.* This project has been funded by the European Union's Horizon 2020 research and innovation programme under grant agreement N° 958927 (Prototype system for a Copernicus $CO_2$ service). CEREA is a member of Institut Pierre-Simon Laplace (IPSL). We would like to acknowledge Tobias Finn for the valuable thought-provoking discussions.





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
