# Peer review of "Deep learning applied to CO2 power plant emissions quantification using simulated satellite images"

_Geoscientific Model Development, 2023_

## Author Comment (AC1)

**Discussion: Deep learning applied to $CO_2$ power plant emissions quantification using simulated satellite images**

Joffrey Dumont Le Brazidec[1], Pierre Vanderbecken[1], Alban Farchi[1], Grégoire Broquet[2],
Gerrit Kuhlmann[3], and Marc Bocquet[1]

[1]CEREA, École des Ponts and EDF R&D, Île-de-France, France
[2]Laboratoire des Sciences du Climat et de l'Environnement, LSCE/IPSL, CEA-CNRS-UVSQ, Université Paris-Saclay, 91198 Gif-sur-Yvette, France
[3]Swiss Federal Laboratories for Materials Science and Technology (Empa), Dübendorf, Switerzland

**Correspondence:** Joffrey Dumont Le Brazidec (joffrey.dumont@enpc.fr)

In the following, the referees comments are in italics and in blue.

**Report 1**

We would like to thank Evan D. Sherwin for the constructive comments and suggestions, which allowed us to clarify several points in the manuscript.

*This paper uses simulated power plant emissions data, generated via the COSMO-GHG model as part of the SMARTCARB project, to simulate carbon dioxide and nitrogen dioxide retrieval via the Copernicus CO2 Monitoring (CO2M) satellite. Developing algorithms for the CO2M satellite is valuable, as the satellite itself will not launch until 2026, limiting possibilities for algorithm development using non-simulated data. The authors focus on the task of quantification, rather than detection, of CO2 emissions from power plants with a known location. The type of power plant is not specified, but presumably they use coal or natural gas.*

We have added "coal-fired" adjective in the dataset presentation section.

*The authors use simulated data from eight power plants, as well as the city of Berlin as the basis for their CO2 quantification efforts, based primarily on a convolutional neural network (CNN) machine learning approach. For each of three selected power plants (Lippendorf, Turow, and Boxberg), the authors train and validate a bespoke version of their CNN model on all power plants but the selected one, which is used as a test dataset. The authors compare quantification error metrics for the baseline CNN with two alternate CNN specifications including the NO2 field and a segmentation map, respectively, as well as with what the authors claim is a standard application of a cross-sectional flux method. The authors apply two interpretability analyses based on analysis of pixel gradients and feature permutation. The results suggest that the CNN is indeed primarily focusing on CO2 emissions from the desired power plant. While acknowledging some of the limitations associated with this simulated data approach, the authors conclude that a CNN-based approach is promising for CO2 quantification with the CO2M satellite once*

*it launches. This paper is a valuable exercise and indeed provides suggestive evidence that a computer vision-based approach such as a CNN can be valuable in CO2 quantification with satellites such as CO2M. However, the approach employed in this paper has several limitations that should be more clearly addressed before it is published.*

Thank you for this accurate summary.

*Detailed comments:*

- *1. The train/validation/test approach taken by the authors does not include a true test set. In standard machine learning, a model is trained and all aspects (including network architecture and hyperparameters) are validated and finalized before any version of the model sees the test set. In the approach employed by the authors, a version of the model is trained and validated on all but the selected power plant, and is then tested on that power plant. However, the fact that this process is repeated at least three times means that any hyperparameter tuning that takes place for the model for the first power plant will translate over to all subsequent models. The authors should clearly acknowledge this limitation and clarify that future work with truly held-out test sets is needed to validate the true performance of such models.*

The experiments are independent: no initial experiment on a power plant provided implicit information for subsequent hyperparameter tuning (like learning rate adjustments) or neural network parameter configurations in other experiments. Furthermore, note that our strategy has involved little to no hyperparameter tuning or validation selection (as we use a rather simple CNN, and ensembling was used instead of a selection between several trained models based on the validation dataset).

- *1. On a related note, it would be valuable to more clearly situate this work in the computer vision remote sensing literature. This is a huge field, one of the most active at the intersection of climate change and AI. How has this work learned from the prior body of accumulated knowledge in this field? What is new in this particular paper? Are there methodological innovations, or does the novelty come solely from the application to CO2M-like data?*

We have added citations to two earlier works including the ones of (Lary et al., 2016; Finch et al., 2021; Jongaramrungruang et al., 2021; Joyce et al., 2023) which are particularly relevant. About the novelty of our work in regard to the existing literature, the reviewer commented that computer vision in remote sensing is an incredibly active field, with an abundance of research making it difficult to be fully aware of all existing works. To our knowledge, a critical methodological innovation of this paper was the choice of the preprocessing layers. The application on CO2M-like data also introduces a number of novel aspects. CO2 data is inherently challenging due to its high noise levels and highly variable plume shapes. Additionally, the incorporation of NO2 data, which is not fully correlated with CO2, and the use of poorly-resolved wind data, add new dimensions to the study.

- *2. The authors train a new, bespoke model for each of the three power plants they focus on in this study. In some cases, they even alter the training dataset to only include other power plants that have emissions similar in magnitude to what they know emissions from the test power plant are. Furthermore, it appears that the test power plant is always in the centre of the scene. Are the authors proposing that a new CNN be trained for every potential CO2 source targeted by*

*CO2M? This seems very inefficient and prone to overfitting. It should be possible to train a generalizable model that both detects and quantifies CO2 emissions at a wide variety of sites. If the authors think this is not the case, they should state this very clearly and explain the rationale for source-specific models, including how they plan to get true emission rate values for all sources for which they plan to train a model.*

As indicated in the paper (and now reiterated as this is a key point), we do not propose to train a new CNN for every potential CO2 source. The proposition of this paper is to train a single CNN model that would be capable of handling all future power plants. This is a methodological paper and what we propose is an "architectural framework" (method + architecture) rather than a ready-to-be-used model. Specifically, our results indicate that the architectural framework is sound. For any given target power plant, we can train a model using the remaining power plants and obtain approximately a $\sim 20\%$ relative error on the target, validating the efficacy of our approach. This serves as strong evidence supporting the architectural framework: if a model trained on 7 power plants can yield accurate results for the eighth (which is unseen), it is highly probable that this model or a model trained on all 8 power plants would likely be accurate for new, unseen power plants.

Our approach is reminiscent of cross-validation techniques. We employ the same methodology, which includes the same CNN model with identical preprocessing layers and hyperparameters, across 9 different scenarios (3 target power plants multiplied by 3 different sets of inputs). In 7 out of these 9 cases, the methodology delivers excellent results without the need for additional tuning. However, in 2 out of the 9 scenarios—specifically when using NO2 or segmentation fields combined with Boxberg—the results were less accurate (but still good and comparable to those of the CSF technique). We traced this to an issue with dataset balance. By removing certain power plants from the training set to achieve better balance in terms of emissions, we were able to improve the performance, thereby indicating that overfitting was the culprit.

Given the importance of this issue, we have expanded the discussion about it in the "Geographical Separation..." section as well as in the conclusion.

– *2. Many power plants in the United States and I think across Europe have continuous emissions monitoring systems that provide ground-truth data. See Cusworth et al. 2021 for more detail. The main value provided by CO2 remote sensing is for CO2 sources that do not have this sort of ground truth.*

We think our paper is consistent with the Copernicus's proposed service to "offer observation-based information to make the assessments more comprehensive and consistent worldwide."

https://climate.copernicus.eu/european-unions-copernicus-programme-planning-monitoring-capacity-anthropogenic-co2-emissions.

One of the objectives of CO2M is to provide independent and consistent estimations, including power plants with ground-truth data.

– *3. The authors acknowledge this to a certain extent, but the structure of the simulated data are likely quite different from true CO2 emissions data. True background noise in greenhouse gas remote sensing is generally not purely Gaussian,*

*but includes significant surface artifacts due to highly reflective/absorptive surface features, as illustrated in Zhang et al. 2023. A model trained only on simulated Gaussian noise may experience difficulty when given more realistic data.*

This paper is about methodological development/exploratory work and is not about the construction of a ready-to-be-used model. It needs to be adapted to be used on real images, as several challenges have not be assessed. This is stated in the Discussions part of the paper: *"In terms of future research, several areas should be explored such as the challenge posed by clouds. In this respect, CNNs can be trained to ignore missing data caused by cloud cover and to make effective use of the available data. Another aspect to consider is the presence of noise in $CO_2M$ data. While Gaussian noise may not pose significant issues, if the satellite noise exhibits structured patterns, it would becomes crucial to develop robust noise modelling techniques to enable CNNs to accurately distinguish and remove such noise. Finally, in real-world applications, training on synthetic datasets and applying the trained models to real datasets may encounter distributional differences. Strategies such as importance weighting, specific data augmentation techniques, transfer learning, or active learning methods may be necessary to account for these differences and ensure reliable performance."* Following your comment, we have added one more mention of systematic error retrieval in the introduction.

- *3. Probably more importantly, the authors use simulated wind data from ERA-5 that appears to be much more uniform and less turbulent than true wind fields. All the plumes shown in this paper appear to be more or less Gaussian, with a little variability in direction (presumably caused by slow changes in wind speed over time).*

The training dataset was generated from high-resolution CO2 simulations with the COSMO-GHG numerical weather prediction model at 1 km resolution (Kuhlmann et al., 2019), which has been shown to realistically capture turbulent plume structures in a recent validation study with measurements (Brunner et al., 2023). It should be noted that at 1-km resolution (CO2M resolution is 2 km), plumes show less turbulent structures than at <100-m as seen in many CH4 measurements (e.g., Joyce et al. (2023)). Nonetheless, many plumes that are assessed by the CNNs do not follow neat Gaussian plume-style dispersion. Figure 2 in the new version of the manuscript is showing a variety of plumes in terms of shape. Furthermore, in our previous paper (Dumont Le Brazidec et al., 2022), more plumes illustrations can be found, many of them not following Gaussian plume-style dispersion (see in particular Fig. 7).

- *4. This raises the related question of limited references to the previous literature in CO2/greenhouse gas remote sensing. CO2M will not be the first CO2 remote sensing instrument. For example, Cusworth et al. 2021 use the PRISMA satellite and the AVIRIS-NG aircraft (based on a spectrometer very similar to the upcoming Carbon Mapper satellite constellation) to detect CO2 emissions from power plants. As you can see from the emissions detected in this paper, they are not really following neat Gaussian plume-style dispersion. More accurately capturing realistic emission shapes will likely require large-eddy simulation, e.g. the approach employed in Gorroño et al. 2023 in the context of satellite-based methane sensing.*

The pixel size in our images is 2km, which is significantly different from the resolutions used in studies like those by Cusworth et al. 2021 or Gorroño et al. 2023. This accounts for the discrepancies between the characteristics of the plumes observed in our work (which are not gaussian) and those described in the mentioned studies. Large Eddy Simulation

(LES) is generally most beneficial for resolving small-scale, turbulent flow structures. In atmospheric sciences or fluid dynamics, it is often used for very high-resolution simulations, often finer than 100 meters. LES seem much more adapted in the case of resolutions such as the one of Cusworth et al. 2021 than in our case (Brunner et al., 2023). With a 2km pixel size, other forms of turbulence modelling are usually adopted.

Length scales have been added on all images.

- *4. The paper mentions the OCO-2 and OCO-3 satellites as already doing CO2 monitoring, but does not include a clear assessment of their CO2 quantification capabilities. How much of an advance would we expect CO2M to be?*
  *It would also be valuable to situate this work in the context of other remote sensing-based GHG monitoring initiatives, such as Climate TRACE 3. GHGSat also has targeted CO2 detection capabilities: https://www.ghgsat.com/en/newsroom/ghgsat-to-launch-worlds-first-commercial-co2-satellite/*

CO2M is designed to provide more comprehensive plume imagery than either OCO-2 or OCO-3. Therefore, it should, in principle, allow for more accurate estimations. This point has been explored in (Danjou, 2022). Our methodology specifically relies on the type of imagery that CO2M will provide, which is not available from OCO-2 or OCO-3. Consequently, applying our convolutional neural network (CNN) approach to the limited data from OCO-2 or OCO-3 (even SAM) would not be appropriate. Some researchers have worked with OCO-2 and OCO-3 data using CSF methods or Gaussian plume models (Nassar et al., 2022). Notably, our work demonstrates that, even on somewhat idealised images, our approach outperforms methods like CSF. The target of OCO-2 is more on natural sources and sinks https://ig3is.wmo.int/en/outcomes/publications/oco-3-mission-overview-science-objectives-and-status while the emphasis of CO2M is on anthropogenic emissions, particularly from point sources like cities and power plants.

Our primary focus in this manuscript is on the CO2M satellite, which remains the leading mission for the scales of interest in our study. While there are numerous other projects and initiatives, including private ones, they often focus on very high-resolution data over small image areas, such as GHGSat. These are not directly comparable to the scales we are examining with CO2M. As for Climate TRACE 3, we found it challenging to understand their methods based on the available information, making it difficult to draw direct comparisons. However, it's worth noting that our general approach could potentially be adapted for finer scales. To do so would require training the model with Large-Eddy Simulation (LES) models, which would be computationally expensive. This could necessitate a reduction in the number of simulated cases to manage computational costs.

- *5. It is difficult to tell how the cross-sectional flux algorithm was implemented and how representative it is of the current standard of practice in the field. Please explain this more clearly, including any ways in which your implementation differs from current standard practice for this method.*

We have revised the manuscript now stating that the CSF method is one of several state-of-the-art techniques that have been recently benchmarked with synthetic CO2M data. The CSF shows similar accuracy than other methods such as Gaussian plume inversion and the light-cross sectional flux method. CSF should therefore be representative for current

standard practice. We have some more details in Section 4.4 describing the CSF method. The implementation is the same as in (Kuhlmann et al., 2020, 2021).

– *6. What do your error metrics mean? Figure 6 and similar figures have no negative values. Are these simply reporting error magnitudes, or are there no underestimates? Absolute value of % error is not a very informative metric here, as an error -99% is quite different than an error of +99%. It would be better to include negative values in error distributions. It would also be useful to compare error metrics to errors achieved in past studies, e.g. Cusworth et al. 2021 and Zhang et al. 2023. It might also be valuable to compare your error metrics with those achieved in satellite-based remote sensing of methane, e.g. Sherwin et al. 2023, but this is not necessary.*

Error metrics are relative and absolute errors:

$$\text{Relative Error} = \frac{|\text{Prediction} - \text{True Value}|}{\text{True Value}} \tag{1}$$

$$\text{Absolute Error} = |\text{Prediction} - \text{True Value}| \tag{2}$$

Following your comment, the absolute component of the relative error has been removed in figures to provide a more insightful metric on the model's tendency to overestimate or underestimate the true value.

About comparisons with errors achieved in past studies, this has been done with the studies of (Kuhlmann et al., 2020, 2021). Beyond these studies, I think comparing error metrics would be misleading: errors can change radically with the dataset considered. Specifically, in our case, in an other study (not yet submitted) performed by two of the co-authors of this paper, the relative error achieved with the cross-sectional fluxes methods was found to be between 26 and 35 %. In this paper, it is found to be close to 40 %. In both cases, the same cross-sectional fluxes method and implementation was used, but a slightly different dataset was used (images from 24-hour per day were used here, while the other study only uses images at CO2M overpass) and caused this $\sim 10\,\%$ difference.

– *7. This study does not appear to include many/any zeroes (instances with zero CO2 flux from the target source). This is a significant limitation and its implications should be discussed in more detail. While this study does not focus on detection, it is presumably still possible to have a false positive (i.e. to estimate nonzero emissions when the power plant is not emitting). This issue is related to the question of class imbalance that the authors note. Class imbalance is very common in computer vision, and a balanced training dataset is not always possible, especially for models aiming to detect/quantify features of multiple sizes. Enforcing an artificially balanced dataset could easily lead to this type of effect if one is not careful. To help clarify these points, I recommend including more detailed summary statistics and/or full time series trend plots of power plant emission rates in the supplementary information. Furthermore, re-training the Boxberg model on a more representative training dataset (presumably after the main Boxberg model had already seen the test data) means that this latter model in particular really does not have a test set in the true sense of the word. This*

*highlights the exploratory nature of this work, which is still valuable, but requires additional testing on independent data to claim external validity.*

This study does not include zeroes but it includes very low emission power plants (from 3 to 40 Mt/yr). Furthermore, plumes from training images are multiplied by a factor of $0.25$ to $2$ (plume scaling) so technically, the CNN is trained on plumes corresponding to emissions of $0.25 \times 3 = 0.75$Mt/yr which is very close to $0$. Therefore, including zero cases is not expected to change our conclusions.

Additionally, CO2M provides an easy indicator for whether a power plant is emitting or not, through local anomalies in NO2 levels. This can serve as a binary switch for activating or not activating the inversion algorithm. Our segmentation algorithm also supports this approach. While the study does not focus on detection per se, the presence of local NO2 anomalies and our segmentation algorithm can help mitigate the risk of false positives, i.e., estimating non-zero emissions when the power plant is not emitting.

Following your comment, plots of the PPs emission rates have been added in the supplementary information.

*Smaller comments:*

– *For simulated satellite images, please include a length scale in kilometers or other appropriate units of distance.*

A length scale in kilometers has been added on Figures 1, 2, 3, 4, 5 (simulated satellite images).

– *Why are hourly emission rates reported in MtCO2/yr instead of an hourly unit?*

The unit MtCO2/yr was the one used in the SMARTCARB dataset. We continued to use it for the sake of practicality.

– *Figure 2: Are the results shown here from training, test, or validation data? Suggest "Targetted" -> "Targeted"*

These results are from the test dataset.

– *L183: When the model takes 4-5 images as input, are these representative of 4-5 separate satellite overpasses? If so, are they from different simulated times (presumably with different emission rates in each)? This should be clarified.*

The 3 to 4 images (not 4 to 5: this was an error that has been corrected) correspond to the $XCO_2$ field and ancillary data such as the winds. We have added details on this in the manuscript.

– *L287: What is the meaning of the "Precisely" in "Precisely, the segmentation model does not discriminate between plume pixels with high amplitude and those with low amplitude."*

"Precisely" is used in a similar sense to "In other words" to emphasise the last part of the previous sentence: "is due to the segmentation model not capturing NO2 or CO2 plume amplitude variations."

– *The results section would be easier to read with a single multi-panel figure for the three power plants side by side, and with one big table instead of one for each power plant. The surrounding text could also be consolidated to be less repetitive.*

We have reorganised the manuscript following your comment. Now section 5.1 is about the model performances, 5.2 is about the two "investigations" (effect of segmentation/NO2 fields, and then overfitting). Thank you, this is indeed better organised now.

– *Figure 10: Why does only Boxberg have a residual density plot? Why does this plot have negative residuals when none of the other density plots do?*

Figure 10 was included as part of the overfitting investigation on Boxberg specifically. This is why it was not shown for Turow/Lippendorf cases. More generally, we have now modified the plot of the relative errors to not be absolute but to show the negative/positive values.

– *In the overfitting section, please include citations about overfitting in ML remote sensing models to support your points here.*

We do not know of an other remote sensing model paper that would be relevant to cite in this section. The reasons why we specify overfitting as the problem here is not linked to remote sensing, satellite data: rather, it is the evolution of the relative errors during training that show us that there is overfitting. We have added a reference to a very good deep learning book (https://d2l.ai/index.html) that tackles overfitting issues.

– *Figure 11: Please include a sentence in the figure caption explaining how it suggests overfitting (presumably the fact that validation error decreases monotonically with number of epochs, while test error does not). Also, make sure to explain what "None", "Segmentation", and "NO2" mean in the figure caption. Also, why are there epochs for the test set? Were the authors applying archived versions of the model from each epoch to the test dataset and computing error?*

We have added a note in the caption to consider your comments. *" Three models are considered: each is trained with the $XCO_2$ field and the winds as inputs. Two of the models additionally assimilate the $NO_2$ field or the predictions of the segmentation model. The validation error decreases monotonically with the number of epochs, while the test error does not, which suggests overfitting of the model."*

We do not keep archived versions of the model. The epoch/error graph in this investigation section has been drawn from a new training of the same model: at each epoch, the model was evaluated on the validation and the test dataset. This was done for investigations purposes (why the original model proposed was not performing well on Boxberg) on overfitting. Given that the architectural framework was adjusted based on the overfitting investigation, this new model needs further validation using an entirely unseen dataset.

– *L360: If you do not include a result in the paper (or at least in the supporting information), then it is best not to reference it in the paper.*

We have removed this result from the paper.

– *Not really clear to me whether there is improvement on the overfitting front around Figure 12. I may be missing something. Where are the results that suggest this?*

We have added the following sentence in the "overfitting investigations" section, to clarify:

*"For example, the median relative error for the CNN with $NO_2$ as additional input is $23.8\%$, comparing to $36.9\%$ in section ..."*

- *When were hyperparameter values set? Before or after any of the models saw test data?*

In section "5.1 Inversion of plumes performance", the choice of the hyperparameters such as learning rate etc has not been tuned based on the test data. In section "5.2.2 Overfitting investigation", the plume scaling hyperparameter specifically as well as the choice of the training dataset was set after seeing test data, as we were investigating why we had overfitting. It is therefore true that this section is purely exploratory.

- *Figure 13: The gradient method shows that the model exclusively focuses on the plume in the centre of the image. This seems like a sign that the model was able to pick up on structured elements of the data it is given, which will not necessarily be present in real satellite data.*

We are not sure to understand or to agree. The plume of interest (the one linked to the target emissions) is always located in the centre of the image. And the model is able to "pick up" the right plume. The model implicitly understands that the plume of interest is the one in the centre of the image. It means that the model successfully understands the relationship between the plume and the targeted emissions. In the case of real images, we can also place the PP in the centre of the image, and if necessary fill in the pixels in the resulting image not measured by the satellite as NaNs values.

- *Table 6: Please explain in the figure caption what "Seg." Means. What are the units of numbers in this table? Are these percent error? What do the colors mean? Simply saying in the caption that the colors will be discussed later is fairly confusing to the reader. Please explain in the figure caption what "Fourth feature" means.*

Seg. is for segmentation. (added in the manuscript). The errors are percent errors (the "degradation in relative error" precisely). We have deleted the colouring as it is forbidden in GMD. Fourth feature has been replaced by "Additional input" and an explanation has been added in the manuscript.

- *In the feature permutation analysis section, please include citations to other studies that do this, or to the method itself. To what extent are the hypotheses listed here supported by the analysis in this paper? How were the colors chosen? Was this arbitrary, or was there a clear method developed in advance of the analysis? Have previous papers done this sort of color-based analysis before?*

A citation to a book describing the feature permutation method has been added in the beginning of the section. The following sentence has also been added *"As input variables used here are not independent, the interpretation of the following permutation analysis should be taken with caution."*

About the colors, these are simply used to clarify the analysis of the table. This was confusing, in any case there seems to be rules in GMD to not allow colouring in tables so we changed.

- *The point about clouds is worth highlighting further. I recommend including some summary statistics of cloudiness in Germany, e.g. from https://earthobservatory.nasa.gov/global-maps/MODAL2_M_CLD_FR.*

Clouds are mentioned several times in the conclusion:

- *"This approach does not account for all the challenges that real satellite images present, specifically issues related to cloud cover and systematic error patterns due to surface reflectance and the aerosol dependency of retrievals."*

- *"Future prospects of the CNN plume inversion method from satellite images encompass the challenges of clouds, cities and real satellite images. Concretely, the method should be able to handle missing data caused by clouds. "*

We prefer not to include statistics of cloudiness in Germany, as the paper is more methodological than on Germany in particular. Furthermore, we do not tackle the challenge of clouds in this paper. However, it should be noted that the SMARTCARB dataset used in this paper includes realistic cloud fraction fields at 1 km model resolution.

- *L459: Suggest "oppurtunities" -> "opportunities"*

Thank you, this typo has been missed.

- *L476: "We demonstrated that the design of a "universal" CNN, trained on a small power plant subset and highly accurate on all of them, is possible." Unless I am missing something, this paper does not do this. As I understand it, the authors train one CNN per target power plant, using other power plants as training and validation data. They appear to use the same network architecture in each case, but these plant-specific models are definitely not a "universal" CNN that is highly accurate on all of them Also, it looks like these bespoke models have significant difficulty if the training dataset includes lower-emitting power plants but the test dataset is a higher-emitting power plant.*

This is a methodological paper and what we propose is an architectural framework rather than a ready-to-be-used model. Specifically, we show that the method/architecture is right since for whatever the target power plant is, we can train a model on the remaining power plants and then achieve a $\sim 20\%$ relative error on the target power plant. This is an argument in favour of the architectural framework: if a model trained on 7 power plants can achieve accurate results on the eighth, then the model trained on all 8 power plants would likely achieve accurate results on new power plants. As it is not a definite proof, we have changed the wording in the conclusion for *"highly suggest that ..."*

About the second part of your comment: we train in total 9 models (3 for each target power plant, and 3 different sets of inputs). Of these 9 models,

- 7 achieve very accurate results with our methodology;
- 2 achieve less accurate results.

Two things should be noted:

- all the models with only $XCO_2$ and winds achieve accurate results. Only models with additional input are achieving less accurate results on Boxberg. This means that the methodology without additional input has proven successful on all power plants;

- in the case of Boxberg, the dataset was significantly unbalanced with a ratio of 5 (lower emission PPs than Boxberg) to 1 (higher emission PP than Boxberg). It was a particularly extreme case.

- *L488: "The training dataset for each CNN is restricted to a dataset conswisting of all other power plants except their target." You mean the training and validation datasets, right?*

  Yes this is what we meant. We have modified the sentence as *"The training/validation dataset for each CNN is restricted to a dataset consisting of all power plants except their target."* for clarity. Thank you.

- *What does the terrain around the power plants look like? Would be good to include satellite images of the three test power plants studied in the main text (together with their surrounding scenes), perhaps including images of the rest of the plants in the SI.*

  Kuhlmann et al. (2019, 2020, 2021) describe these synthetic satellite observations in depth and provide numerous additional details.

- *Would be good to have numbers in the abstract, e.g. the error profile of the best-performing method*

  This is a very good idea. We have added *"and a relative error of $20\%$ when only the $XCO_2$ and wind fields are used as inputs."* in the abstract. Thank you for the suggestion.

Many thanks to this reviewer for his time and long, complete, and precise review of our work. It allowed to highlight several points of this paper that were probably unclear in the current state and we think it really improves the overall clarity. We have added a note of thanks in the acknowledgments section of the paper.

**References**

Brunner, D., Kuhlmann, G., Henne, S., Koene, E., Kern, B., Wolff, S., Voigt, C., Jöckel, P., Kiemle, C., Roiger, A., Fiehn, A., Krautwurst, S., Gerilowski, K., Bovensmann, H., Borchardt, J., Galkowski, M., Gerbig, C., Marshall, J., Klonecki, A., Prunet, P., Hanfland, R., Pattantyús-Ábrahám, M., Wyszogrodzki, A., and Fix, A.: Evaluation of simulated $CO_2$ power plant plumes from six high-resolution atmospheric transport models, Atmospheric Chemistry and Physics, 23, 2699–2728, https://doi.org/10.5194/acp-23-2699-2023, publisher: Copernicus GmbH, 2023.

Danjou, A.: Émissions de CO2 estimées par données satellitaires sur les villes à forte croissance démographique, phdthesis, Université Paris-Saclay, https://theses.hal.science/tel-04042933, 2022.

Dumont Le Brazidec, J., Vanderbecken, P., Farchi, A., Bocquet, M., Lian, J., Broquet, G., Kuhlmann, G., Danjou, A., and Lauvaux, T.: Segmentation of $XCO_2$ images with deep learning: application to synthetic plumes from cities and power plants, Geoscientific Model Development Discussions, pp. 1–29, https://doi.org/10.5194/gmd-2022-288, publisher: Copernicus GmbH, 2022.

Finch, D., Palmer, P., and Zhang, T.: Automated detection of atmospheric $NO_2$ plumes from satellite data: a tool to help infer anthropogenic combustion emissions, Atmos. Meas. Tech., pp. 1–21, https://doi.org/10.5194/amt-2021-177, 2021.

Jongaramrungruang, S., Matheou, G., Thorpe, A. K., Zeng, Z.-C., and Frankenberg, C.: Remote sensing of methane plumes: instrument tradeoff analysis for detecting and quantifying local sources at global scale, Atmospheric Measurement Techniques, 14, 7999–8017, https://doi.org/10.5194/amt-14-7999-2021, publisher: Copernicus GmbH, 2021.

Joyce, P., Ruiz Villena, C., Huang, Y., Webb, A., Gloor, M., Wagner, F. H., Chipperfield, M. P., Barrio Guilló, R., Wilson, C., and Boesch, H.: Using a deep neural network to detect methane point sources and quantify emissions from PRISMA hyperspectral satellite images, Atmospheric Measurement Techniques, 16, 2627–2640, https://doi.org/10.5194/amt-16-2627-2023, publisher: Copernicus GmbH, 2023.

Kuhlmann, G., Broquet, G., Marshall, J., Clément, V., Löscher, A., Meijer, Y., and Brunner, D.: Detectability of $CO_2$ emission plumes of cities and power plants with the Copernicus Anthropogenic $CO_2$ Monitoring (CO2M) mission, Atmos. Meas. Tech., 12, 6695–6719, https://doi.org/10.5194/amt-12-6695-2019, 2019.

Kuhlmann, G., Brunner, D., Broquet, G., and Meijer, Y.: Quantifying $CO_2$ emissions of a city with the Copernicus Anthropogenic $CO_2$ Monitoring satellite mission, Atmos. Meas. Tech., 13, 6733–6754, https://doi.org/10.5194/amt-13-6733-2020, 2020.

Kuhlmann, G., Henne, S., Meijer, Y., and Brunner, D.: Quantifying CO2 Emissions of Power Plants With CO2 and NO2 Imaging Satellites, Front. remote sens., 2, https://www.frontiersin.org/article/10.3389/frsen.2021.689838, 2021.

Lary, D. J., Alavi, A. H., Gandomi, A. H., and Walker, A. L.: Machine learning in geosciences and remote sensing, Geoscience Frontiers, 7, 3–10, https://doi.org/10.1016/j.gsf.2015.07.003, 2016.

Nassar, R., Moeini, O., Mastrogiacomo, J.-P., O'Dell, C. W., Nelson, R. R., Kiel, M., Chatterjee, A., Eldering, A., and Crisp, D.: Tracking CO2 emission reductions from space: A case study at Europe's largest fossil fuel power plant, Frontiers in Remote Sensing, 3, https://www.frontiersin.org/articles/10.3389/frsen.2022.1028240, 2022.

---

## Author Comment (AC2)

**Discussion: Deep learning applied to $CO_2$ power plant emissions quantification using simulated satellite images**

Joffrey Dumont Le Brazidec[1], Pierre Vanderbecken[1], Alban Farchi[1], Grégoire Broquet[2], Gerrit Kuhlmann[3], and Marc Bocquet[1]

[1]CEREA, École des Ponts and EDF R&D, Île-de-France, France
[2]Laboratoire des Sciences du Climat et de l'Environnement, LSCE/IPSL, CEA-CNRS-UVSQ, Université Paris-Saclay, 91198 Gif-sur-Yvette, France
[3]Swiss Federal Laboratories for Materials Science and Technology (Empa), Dübendorf, Switzerland

**Correspondence:** Joffrey Dumont Le Brazidec (joffrey.dumont@enpc.fr)

In the following, the referees comments are in italics and in blue.

**Report 2**

We would like to thank the anonymous Referee 2 for her/his technical comments and suggestions on improving the manuscript.

*The authors present a method to constrain power plant's CO2 emissions with a deep learning model. The model was trained and tested with simulated CO2 concentrations from SMARTCARB simulations. The analysis demonstrated the superior performance of the deep model to contain the low signal-to-noise ratio issue and is significantly better than the traditional CSF method. The topic is interesting, and the method is helpful for better inversion of CO2 emission in the future. However, there are still issues that need to be addressed before the paper can be published, particularly, an extended comparative analysis is suggested to provide a better understanding of the potential advantages and limitations of this method.*

*Main comments:*

– *Line 42, the authors indicate that the purpose of this study is to address the second and third problems in the CO2 inversions: 2) the low signal-to-noise ratio and 3) the uncertainty in the transport and dispersion processes. I agree that the results demonstrated the remarkable ability to address the low signal-to-noise ratio issue, however, it is unclear how this analysis can address the issue of uncertainty in the transport and dispersion processes because the training and test are both based on SMARTCARB simulations by assuming no systematic errors in simulations.*

The "uncertainty in transport and dispersion processes" refers to the challenge of accurately estimating emissions from a well-characterised plume. Traditional methods like cross-sectional fluxes are highly sensitive to small errors in wind assessment, which can lead to significant errors in emission estimates. Our Convolutional Neural Network (CNN) approach, however, appears to estimate emissions accurately even when there are potential errors in the wind fields used as

inputs. These wind fields are not the same as those used to compute the SMARTCARB simulations, indicating robustness against such uncertainties.

We acknowledge the concern about potential systematic errors in the model, which could bias the training process. For instance, if real-world plumes are systematically different in some way not captured by the model (e.g., thinner due to lack of numerical dispersion), this could introduce bias. However, based on existing literature, we believe that such biases are not significant enough to make the SMARTCARB simulations fundamentally disconnected from reality. This is supported by publications like (Brunner et al., 2023), which suggest that mesoscale plume behaviours are generally well-represented in models.

Furthermore, classic inversion methods which rely on transport models typically compare the observed plume at time $t$ with a model simulation at the same time. Should the model inaccurately represent the plume at this time, this can lead to substantial errors. However, the CNN model implicitly compares the observed plume at time $t$ with a training dataset comprising multiple plume scenarios, thus providing a broader scope to mitigate errors that a single model simulation might introduce.

– *Sections 5.1-5.3: Here the model performance is demonstrated for Lippendorf, Turow, and Boxberg individually. It is suggested to demonstrate the performances of all PPs, including the city of Berlin, and provide a comparative discussion for the model performance over these PPs to investigate the possible consistency and discrepancy, which can provide a better understanding of the potential advantages and limitations of this method.*

Implementing this suggestion would entail a significant amount of work. Choosing three target power plants represented a balance between adequate performance evaluation and computational efficiency. For each PP, we study three models: one with XCO2, winds; one with XCO2, winds, NO2; one with XCO2, winds, segmentation fields. For each model we need ~10 CNN training runs (as the final model is an ensemble of models). So each study on a new PP requires near 30 CNN training runs. And therefore an extension of our work on the 5 other sources would represent 150 CNN training runs and other costs due to the constitution of the datasets, management of the results, etc. This would imply considerably more work and computation time.

We knew about these costs and the limits. We therefore carefully selected the target power plants, based on several criteria.

– a first criterion is that we wanted a variety of power plants, in terms of emissions (one low, one average, and one high emission power plant)

– a second criterion is that we did not want power plants on the borders of the SMARTCARB domain. As you can see on Figure 1, Patnow or Opole are close to the borders and are subject to border conditions.

– a third criterion was to not take into account power plants on the extrema in terms of emission rates (Boxberg was considered rather than Jänschwalde because Jänschwalde was the highest emission PP of our dataset and may fall "out-of-training" distribution).

[Figure]

**Figure 1.** XCO$_2$ concentration map with the locations of Berlin and each considered PP within the complete SMARTCARB domain. The map consists only of the concentrations stemming from the major anthropogenic sources. Furthermore, to enhance plume visibility, as fluxes of power plants such as Jänschwalde are vastly superior to other fluxes, concentrations exceeding 2 ppmv have been capped at 2 ppmv.

– we also considered images with multiple power plants on the same image (such as Boxberg or Turow)

This discussion has been added in the manuscript (section: "Geographical separation between ...").

Finally, we can expect that the current CNN does not generalise well on Berlin for the following reason: we have no cities (apart from Berlin) in our dataset. A CNN used to predict the emissions of Berlin would be only trained on PPs which have very different plumes from those of a city.

– *Sections 5.1.1, 5.2.1 and 5.3.1 show the model performances, while Section 5.1.2 shows the effect of segmentation and NO2 fields and Section 5.3.2 shows overfitting investigation. The organization of these sections is orderless and needs to be improved.*

We have reorganised the manuscript following your comment. Now section 5.1 is about the model performances, 5.2 is about the two "investigations" (effect of segmentation/NO2 fields, and then overfitting). Thank you for this suggestion.

*Technical comments:*

– *Abstract: The abbreviations, such as CO2M, CO2 and NO2, should be defined.*

The abbreviations have been defined in the abstract.

– *It could be better to list the first, second and third problems more clearly, for example, 1); 2) and 3), otherwise, readers have to check Lines 34-41 carefully to determine which problems are the second and third.*

Your suggestion has been added to clarify what are the three problems. Thank you.

– *Figure 2: How is the targeted plume obtained?*

In the SMARTCARB dataset, the full CO2 field is made by summing up several components including the background and the major anthropogenic plumes. The targetted plume field is then obtained from the major anthropogenic plumes field after application of a pixel-wise weighting function described in (Dumont Le Brazidec et al., 2022).

– *Line 131: The title of Section 4 should be "Deep learning method for the inversion of XCO2".*

Thanks. We had missed this typo.

– *Table 1: It would be better to have a map to show the locations of these PPs.*

This has been added just next to Table 1. It was indeed missing, thank you.

– *Fig 3: The model input features are the XCO2 image, u-wind, v-wind and additional NO2 field or segregation model output contour. Figure 3 is not fully drawn and may be misinterpreted by the reader as inputting only the four features in the figure.*

It was indeed misleading. We have added the v-wind field in the figure and a note in the title to avoid misinterpretations.

– *Line 159-164: How was the range of these factors determined?*

Plume scaling factors ($p$ and $a$) were chosen so that enhanced plumes are still in the range of "possible" plumes. $a$ range is bigger than $p$ range because the alternate anthropogenic fluxes field is supposed to be composed of lower plumes than the "major" anthropogenic fluxes field. Similarly, the background modifier $b$ was chosen as the standard deviation of an average background so that enhanced background are still in the range of "possible" backgrounds. We have added a sentence about this in the manuscript.

– *Line 169: What kind of standardization is used?*

We use "Z-score normalization" (scaling the values of a feature to a mean of 0 and a std of 1). It is now precised in the manuscript.

– *It could be better to provide a brief explanation or definition for the kernel density (e.g., Fig. 6).*

The sentence: *"KDE is a non-parametric statistical technique that estimates the probability density function of a continuous random variable by smoothing its observed data points using a kernel function."* has been added in the manuscript. Here KDEs are preferred to histograms solely for visibility reasons.

Thank you very much for all these very clear and helpful remarks. We have added a note of thanks in the acknowledgments section of the paper.

Additionally, we have made available a cleaner version of the code on Zenodo and GitHub.

**References**

Brunner, D., Kuhlmann, G., Henne, S., Koene, E., Kern, B., Wolff, S., Voigt, C., Jöckel, P., Kiemle, C., Roiger, A., Fiehn, A., Krautwurst, S., Gerilowski, K., Bovensmann, H., Borchardt, J., Galkowski, M., Gerbig, C., Marshall, J., Klonecki, A., Prunet, P., Hanfland, R., Pattantyús-Ábrahám, M., Wyszogrodzki, A., and Fix, A.: Evaluation of simulated $CO_2$ power plant plumes from six high-resolution atmospheric transport models, Atmospheric Chemistry and Physics, 23, 2699–2728, https://doi.org/10.5194/acp-23-2699-2023, publisher: Copernicus GmbH, 2023.

Dumont Le Brazidec, J., Vanderbecken, P., Farchi, A., Bocquet, M., Lian, J., Broquet, G., Kuhlmann, G., Danjou, A., and Lauvaux, T.: Segmentation of $XCO_2$ images with deep learning: application to synthetic plumes from cities and power plants, Geoscientific Model Development Discussions, pp. 1–29, https://doi.org/10.5194/gmd-2022-288, publisher: Copernicus GmbH, 2022.

---

## Author Response (AR2)

**Discussion: Deep learning applied to $CO_2$ power plant emissions quantification using simulated satellite images**

Joffrey Dumont Le Brazidec[1], Pierre Vanderbecken[1], Alban Farchi[1], Grégoire Broquet[2], Gerrit Kuhlmann[3], and Marc Bocquet[1]

[1]CEREA, École des Ponts and EDF R&D, Île-de-France, France
[2]Laboratoire des Sciences du Climat et de l'Environnement, LSCE/IPSL, CEA-CNRS-UVSQ, Université Paris-Saclay, 91198 Gif-sur-Yvette, France
[3]Swiss Federal Laboratories for Materials Science and Technology (Empa), Dübendorf, Switzerland

**Correspondence:** Joffrey Dumont Le Brazidec (joffrey.dumont@enpc.fr)

In the following, the referees comments are in italics and in blue.

**Report 1**

We would like to thank again Evan D. Sherwin for his comments or requests for clarification, we hope the latest version of this manuscript is even clearer.

*The revised manuscript is significantly improved and largely addresses my comments. I still recommend the following relatively minor revisions before acceptance. As a general point, it is helpful to reviewers if the response document reproduces key changes to the text (or relevant pre-existing parts of the text). For each reviewer comment, the authors should generally point to one or the other (or both).*

*Detailed comments:*

- *1.1: The response says "experiments are independent" and that no initial experiment provided implicit info for model tuning. While I am sure this is true, the authors should state that in all cases in this study, there was no independent test set that had not been used by any other models used in this study by the authors. The authors are then welcome to make a plausibility case along the lines they do in this response as to why the approach they took may be justified in this instance.*

To account for this suggestion, we have added "It is important to note that while the test dataset from one experiment appears in the training dataset of another, each experiment was conducted independently. The model tuning was not influenced by the results obtained with the test datasets" in section "4.3 Geographical separation between the training and test datasets".

- *1.2: Computer vision for remote sensing is indeed a very active field, but if the authors want to make the case that the choice of preprocessing layers is a critical methodological innovation, citing two other papers does not really*

*seem like enough. Incorporation of NO2 data is probably relatively novel, but I recommend citing other papers that attempt CO2 remote sensing and do not use NO2 data (they do not need to be computer vision/ML papers). My understanding is that plenty of studies use the sort of poorly-resolved wind reanalysis data used in this study for emissions estimation. Please cite more additional studies than just this one, but here is an example worth mentioning: Kumar et al. 2023, computer vision-based remote sensing of methane: https://openaccess.thecvf.com/content/CVPR2023/html/ Kumar_MethaneMapper_Spectral_Absorption_Aware_Hyperspectral_Transformer_for_Methane_Detection_CVPR_2023_ paper.html*

NO2 data (for prediction of the position of the plume) or winds data are indeed already used in other papers (cited in our manuscript). What we were referring as "critical methodological innovation" was the data generation process choices and set-up (precisely: the succession of the six steps in section 4.2.1 Description of the preprocessing layers) rather than the choice of the fields used as inputs. In other words, what is critical is the data augmentation set-up to enhance the relatively small dataset.

We actually show in the results section that the addition of NO2 or not has a small impact, and that winds are not critical input fields (although they help in the inversion).

The paper of Kumar et al. 2023 is interesting and we have added a citation in the manuscript. But this is a work on methane (which does not have the low signal-to-noise ratio issue of $CO_2$ satellite images, among other important differences) and at a much higher resolution (1.5m/pixel). We have favored citation of works on $CO_2$, preferably at similar resolutions (CO2M).

We have also added a citation in the introduction to (Santaren et al., 2024) which analyses $CO_2$ plume inversion methods (Cross sectional fluxes, Gaussian, ...) on the SMARTCARB dataset.

Please also note that 4 citations had been added during the first review round (rather than 2).

– *2.1: If the goal of this paper is to train a single CNN for all power plants, I recommend stating this clearly early on in the manuscript. The fact that the authors removed some power plants from the training set to improve model performance in the two cases where the model was not as good a fit should be clearly identified in the manuscript by the authors as a point of caution for future model developers aiming to produce a more generalizable model. It will not be possible to enforce this type of balance in training sets for a model that works for all (or most) power plants and/or cities.*

The goal of this paper is to prove that training a (single) CNN for all power plants is worthwhile: the focus is on the strategy rather than on the final model. We agree with the need for caution, and we have incorporated a statement to this effect in the Discussions and Limitations section: "Secondly, the study emphasises the importance of a balanced dataset, as highlighted by the section .., and the need to be able to identify and address potential overfitting issues."

Furthermore, we have modified a sentence in the introduction to state more clearly the objective: "In this paper, we develop a strategy employing convolutional neural networks (CNN) to estimate the emission fluxes from a plume in a pseudo $XCO_2$ image. "

Finally, in the last revision, the sentence "The objective behind creating these models is to demonstrate the effectiveness of the architectural framework, laying the groundwork for a universal model based on this architecture and capable of generalising across future PPs." in section "Geographical separation between the training and test datasets" has been added to make the objective clear.

– *2.2: Please add text to the manuscript to address the point on the existence of continuous emissions monitoring systems in the US and probably the EU, or indicate where in the manuscript it is already addressed. I think this point is worth highlighting, since these remote sensing systems will likely add the most value outside the US and EU.*

It is not clear to us how the paper (Cusworth et al., 2021) (mentioned in the previous review round) provides a continuous emissions monitoring system. The estimations of (Cusworth et al., 2021) are estimations from other types of observations that those obtained with CO2M. Furthermore, in the abstract of (Cusworth et al., 2021), it is mentioned: "We highlight four examples of coal-fired power plants in India, Poland, and South Korea, where we quantify significant carbon dioxide emissions from power plants where limited public emissions data exist" so this paper does not target specifically the US or the EU.

To cite this interesting paper, we have added the following text in the introduction: "and complement other estimations (Cusworth et al., 2021)."

– *4. Please include language in the manuscript summarizing the points in this response about other CO2-sensing satellites.*

From our answer in the previous review round, the main points were:

1. CO2M aims to provide more comprehensive plume imagery than OCO-2 or OCO-3, potentially allowing for more accurate estimations.

2. The methodology discussed relies specifically on the type of imagery CO2M will provide, which differs from that of OCO-2 or OCO-3.

3. The research demonstrates that the proposed approach outperforms other methods, like CSF, on idealised images.

4. OCO-2 focuses more on natural CO2 sources and sinks, while CO2M targets anthropogenic emissions from point sources.

5. The study primarily focuses on CO2M, which aligns with the scales of interest, unlike other high-resolution over small image areas projects.

6. Adapting the approach for finer scales would require computationally expensive Large-Eddy Simulation (LES) models and might necessitate reducing the number of simulated cases.

Points 1-4 are already addressed in the introduction or conclusion. We have added a mention to points 5 and 6 in the "Discussions and Limitations" section: "Finally, the method could be modified to extract CO2 emissions from plume imagery at more detailed scales, necessitating the use of resource-intensive Large-Eddy Simulation (LES) models."

- *7. Please include discussion in the main manuscript along the lines of your response on the absence of zeros in the datasets, the possibility of false positives, and your assessment of the implications of this. Old Figure 2 (different figure number now): Please clarify in the figure caption that these are results from the test dataset. Old L183: Where in the manuscript were these details added? Would be helpful to reproduce the text in the response document. The new organization is now easier to follow. Thank you.*

  In section "Discussions and limitations", we have included: "A last limitation is the absence of zero-emission source in the dataset. However, the inclusion in the training dataset of very low emission power plants and of a plume scaling approach generating near-zero emission plumes indicates that incorporating zero emission cases would likely not markedly change the outcomes."

  Clarifications on figure caption (old figure 2) have been added in the figure caption.

  It was added at line 191 (new). "The chosen model takes 3 to 4 images of $64 \times 64$ pixels as input (which correspond to the $XCO_2$ field and ancillary data such as the winds)."

- *Old Figure 9: Please explain this to the reader in the manuscript. Also, please clarify in this figure that units are in %. Timing of hyperparameter value selection: Please summarize these points explicitly in the main manuscript*

  We are not entirely sure what should be explained from Old figure 9 in the manuscript: the old Figure 9 is now included in the "new" Figure 7. This figure is described and analysed in the manuscript (the whole text of section 5.1 describes Figure 7 and the following table). In Figure 7, only left column is in percents, which is indicated.

  The timing of hyperparameter value selection is now mentioned in section "Geographical separation between the training and test datasets": "(or the hyperparameter selection such as the learning rate that were selected prior to the training of the models)".

- *Old Figure 13: Please mention in the main manuscript that the modeling approach applied requires that CO2 plumes be located at the center of an image, and include a sentence or two of commentary summarizing the points you make in this response.*

  We have added the following text: "In conclusion, the model consistently identifies the target emission plume situated at the image's centre, indicating it implicitly understands the relationship between the plume and targeted emissions. " in section "Gradient-based study of the pixels".

  To indicate that we work on CO2 plumes located at the center of an image, we have modified a sentence in the section "Inversion based on supervised learning" to make it "The inverse problem addressed here is the estimation of the $CO_2$ emissions accountable for the central hotspot plume observed in a given $XCO_2$ field image."

Again, many thanks to the Reviewer and the Editor for their time.

**Report 2**

No additional comments

**References**

Cusworth, D. H., Duren, R. M., Thorpe, A. K., Eastwood, M. L., Green, R. O., Dennison, P. E., Frankenberg, C., Heckler, J. W., As-
ner, G. P., and Miller, C. E.: Quantifying Global Power Plant Carbon Dioxide Emissions With Imaging Spectroscopy, AGU Advances,
2, e2020AV000 350, https://doi.org/10.1029/2020AV000350, _eprint: https://onlinelibrary.wiley.com/doi/pdf/10.1029/2020AV000350,
2021.

Santaren, D., Hakkarainen, J., Kuhlmann, G., Koene, E., Chevallier, F., Ialongo, I., Lindqvist, H., Nurmela, J., Tamminen, J., Amoros, L.,
Brunner, D., and Broquet, G.: Benchmarking data-driven inversion methods for the estimation of local $CO_2$ emissions from $XCO_2$ and
$NO_2$ satellite images, Atmospheric Measurement Techniques Discussions, pp. 1–52, https://doi.org/10.5194/amt-2023-241, publisher:
Copernicus GmbH, 2024.